# Holobiome Harmony: Linking Environmental Sustainability, Agriculture, and Human Health for a Thriving Planet and One Health

**DOI:** 10.3390/microorganisms13030514

**Published:** 2025-02-26

**Authors:** Gissel García, Martha Carlin, Raul de Jesus Cano

**Affiliations:** 1Pathology Department, Hospital Hermanos Ameijeiras, La Habana 10400, Cuba; gisselgarcia2805@gmail.com; 2The BioCollective, LLC, Aurora, CO 80216, USA; martha.carlin@thebiocollective.com; 3Biological Sciences Department, California Polytechnic State University, San Luis Obispo, CA 93407, USA; 4Chauvell, LLC, San Luis Obispo, CA 93401, USA

**Keywords:** holobiome, microbiome, probiotics, sustainable agriculture, soil health, gut microbiota, microbial diversity, climate resilience, artificial intelligence (AI), ecosystem balance

## Abstract

The holobiome is an interconnected network of microbial ecosystems spanning soil, plants, animals, humans, and the environment. Microbial interactions drive nutrient cycling, pathogen suppression, and climate regulation. Soil microbiomes facilitate carbon sequestration and enhance soil fertility, while marine microbiomes contribute to carbon capture and climate stability. However, industrial agriculture, extensive herbicide use, antibiotic overuse, and climate change threaten microbial diversity, leading to ecosystem and health disruptions. Probiotic interventions help to restore microbial balance. In human health, probiotics support gut microbiota diversity, reduce inflammation, and regulate metabolism. In agriculture, soil probiotics enhance microbial diversity, improve nutrient cycling, and degrade contaminants, increasing crop yields and soil health. Case studies show that microbial inoculants effectively remediate degraded soils and enhance nutrient uptake. Artificial intelligence is transforming microbiome research by enabling predictive modeling, precision probiotic design, and microbial consortia optimization. Interdisciplinary collaboration and supportive policies are essential for restoring microbial equilibria, ensuring ecosystem resilience, and promoting long-term sustainability. The integration of artificial intelligence, clinical research, and sustainable practices is crucial for advancing holobiome science. The holobiome framework underscores the need for interdisciplinary collaboration to address global challenges, bridging environmental sustainability, agriculture, and public health for a resilient future.

## 1. Introduction

### 1.1. Background

The term “holobiome” originates from the prefix “holo-” (ὅλος), meaning “whole” or “entire”, and “biome”, which refers to a community of living organisms. It defines an ecological framework in which a host organism and its associated microorganisms function as a unified system.

The concept of the holobiome has evolved significantly over time. The term “holobiont” was first introduced by the German theoretical biologist Adolf Meyer-Abich in 1943 [1]. However, it was later popularized by Dr. Lynn Margulis in 1991 [1]. The term “holobiont” refers to the assemblage of a host and its associated microorganisms, forming a single ecological unit. Interest in holobiome studies grew with advances in metagenomics and high-throughput sequencing technologies, which allowed researchers to explore the complex interactions between hosts and their microbial communities. The hologenome theory of evolution, proposed by Eugene Rosenberg and Ilana Zilber-Rosenberg in 2008 [2], further emphasized the importance of these interactions, suggesting that the holobiont and its collective genome, the hologenome, function as a distinct biological entity subject to natural selection. This growing interest has led to a better understanding of the role of microbiomes in health, disease, and evolution.

Closely related to the concept of the holobiont, which describes the host–microbe collective, the holobiome extends beyond individual hosts, encompassing interconnected microbial ecosystems across soil, plants, animals, humans, and the broader environment. This integrated perspective highlights the dynamic interactions that sustain biological and ecological balance across multiple scales [3]. It recognizes the centrality of microbial life in supporting biological and ecological functions, emphasizing that the health of one ecosystem cannot be isolated from others. Microbes, which include bacteria, archaea, fungi, and viruses, are fundamental to life on Earth, driving critical processes such as nutrient cycling, organic matter decomposition, and the regulation of biogeochemical cycles [4,5]. The concept of the holobiome encapsulates this interconnectedness, highlighting how microbial ecosystems interact to maintain global stability and health.

At its core, the holobiome underscores the intricate relationship between microbial diversity and ecosystem resilience. Soil microbes, for instance, facilitate nitrogen fixation, carbon sequestration, and the decomposition of organic matter, processes that are vital for plant growth and agricultural productivity [6]. Similarly, plant-associated microbes in the rhizosphere protect against pathogens and enhance nutrient uptake, directly influencing food production and quality [7]. These benefits extend to human health, as the human gut microbiome—a diverse community of trillions of microbes—plays a pivotal role in digestion, immunity, and the modulation of systemic inflammation [8]. Thus, the holobiome represents not just an ecological construct, but also a framework for understanding the symbiotic interactions that sustain life on Earth.

Achieving a resilient future through holobiome applications requires sustained investment in microbiome research to elucidate the intricate host–microbe and microbe–microbe interactions that drive holobiome functionality. Advancing this field necessitates interdisciplinary collaboration among microbiologists, ecologists, agricultural scientists, policymakers, and industry stakeholders to translate fundamental discoveries into scalable, evidence-based applications [9].

The integration of holobiome principles into sustainable agricultural and environmental management practices can enhance ecosystem stability, improve soil and plant health, and mitigate the impacts of environmental stressors. Furthermore, public engagement and knowledge dissemination are critical in fostering the widespread adoption of holobiome-based interventions and ensuring policy alignment with emerging scientific insights [10].

By leveraging the functional potential of holobiomes, we can develop robust, adaptive strategies that promote long-term ecological and human resilience in the face of global environmental and health challenges [11].

Microbial interdependence is a defining feature of the holobiome, where the health and functionality of one microbiome influence others. Soil microbes, for example, are essential for sustaining plant growth by breaking down organic matter and cycling nutrients like nitrogen, phosphorus, and potassium [12,13,14]. Plants, in turn, provide energy to these microbes through root exudates, creating a feedback loop that enhances soil fertility and crop resilience. This relationship is disrupted by practices such as excessive tillage, monoculture farming, and the overuse of chemical fertilizers, which reduce microbial diversity and impair ecosystem services [15,16,17].

Human health is also intrinsically linked to environmental microbiomes. The gut microbiome, often referred to as a “second genome”, directly interacts with the microbiomes of the food we consume, which are shaped by agricultural practices and soil health [18,19,20]. For instance, the microbial composition of organically grown produce differs significantly from conventionally grown produce treated with synthetic pesticides, potentially influencing gut microbial diversity and overall health outcomes [21].

The gut microbiome plays a crucial role in human health, influencing not only microbial diversity, but also a wide range of physiological processes [22]. Dysbiosis, or an imbalance in the gut microbial composition, has been associated with several chronic conditions, including an increased risk of inflammatory bowel disease (IBD), metabolic disorders such as type 2 diabetes, obesity, and even neurological conditions like depression and cognitive decline [23,24,25]. The intricate relationship between gut microbiota and host metabolism underscores the need to better understand microbial contributions to health outcomes and potential interventions for restoring microbial balance.

Furthermore, exposure to diverse soil and environmental microbiomes during early life has been shown to enhance immune development and reduce the risk of chronic conditions such as allergies and asthma [26,27]. Conversely, disruptions such as gut dysbiosis—a microbial imbalance in the gut—can exacerbate inflammatory diseases, metabolic disorders, and even mental health conditions, underscoring the systemic impact of microbiome perturbations [28,29].

The interdependence of these microbiomes becomes particularly evident during disruptions. For example, soil degradation caused by glyphosate contamination not only reduces soil microbial diversity, but also affects crop nutrient density, indirectly impacting human health through nutrient-deficient diets [30,31,32,33]. Similarly, antibiotic use in livestock alters animal microbiomes, and residues entering soil and water systems propagate dysbiosis across multiple domains [34,35,36]. These cascading effects demonstrate the fragility of the holobiome and the urgent need for interventions that restore microbial balance at every level.

Holobiome research faces several significant challenges. The complexity of microbial interactions makes it difficult to establish universal models, as holobiomes consist of diverse microbial communities that interact dynamically with their host and environment. These interactions are influenced by diet, genetics, and external stressors, making their study highly context-dependent [37].

Technological limitations also present obstacles. While high-throughput sequencing and multi-omics approaches have advanced microbiome characterization, challenges remain in resolving microbial composition at the strain level, understanding functional roles, and detecting low-abundance species critical for holobiome function.

Additionally, environmental variability complicates standardization. Microbiome composition and function fluctuate due to factors like climate, soil conditions, pollutants, and host physiology, making it difficult to develop consistent applications across different ecosystems [38].

Significant knowledge gaps regarding microbe–host interactions persist, particularly in understanding how microbiomes influence host metabolism, immunity, and overall health. Addressing these gaps is crucial for advancing microbiome-based interventions. Furthermore, ethical and regulatory concerns must be considered, especially regarding potential ecological consequences, gene transfer risks, and the long-term stability of engineered microbial communities [39,40].

Despite these challenges, holobiome research holds immense promise. In agriculture, microbiome-based strategies can improve crop resilience, enhance nutrient uptake, and reduce reliance on chemical inputs. In climate change mitigation, microbial communities play a key role in carbon and nitrogen cycling, offering potential solutions for carbon sequestration and soil restoration [9,41].

For human health, holobiome research enables personalized microbiome-based interventions that can enhance gut health, modulate immunity, and improve disease prevention [42]. In environmental conservation, microbiome applications can restore biodiversity, aid in pollutant bioremediation, and stabilize ecosystems [43]. Finally, advances in biotechnology could lead to next-generation biofertilizers, biopesticides, and bio-based alternatives to chemical inputs, promoting sustainable agricultural and environmental practices [44].

By addressing these challenges and harnessing the potential of holobiome research, we can drive innovations that contribute to a more resilient and sustainable future.

Figure 1 provides a conceptual overview of the holobiome’s interconnected roles in human health, agriculture, and environmental sustainability. The illustration highlights the dynamic interactions between microbial communities, plant and soil ecosystems, and host physiology, emphasizing the influence of microbiome-derived metabolites such as short-chain fatty acids (SCFAs) on systemic functions. This integrative framework underscores the holobiome’s potential in advancing sustainable health and ecological resilience.

This figure provides a visually integrated representation of the holobiome concept, illustrating the interconnectedness of soil health, human health, and environmental sustainability. At its center, a human figure is surrounded by microbial networks, symbolizing the gut microbiome’s essential role in maintaining health and its dynamic relationship with the surrounding environment. Adjacent to the human, thriving plants with extensive root systems interact with diverse soil microbial communities, emphasizing the microbiome’s critical role in nutrient cycling, plant resilience, and overall ecosystem stability. This figure was generated using DALL·E, an AI-driven image generation tool, based on a conceptual framework integrating microbiome science, ecological interactions, and metabolic functions (https://openai.com/dall-e, accessed on 21 January 2025). The design elements were selected to depict the synergistic relationships within holobiomes, highlighting the flow of biological information across different levels of life.

### 1.2. Scope and Objectives

This paper explores the interconnected roles of microbial ecosystems within the holobiome, focusing on how agricultural practices, clinical studies, and rationally designed probiotics contribute to restoring and enhancing microbial health. The holobiome framework recognizes that microbial communities across different environments—soil, plants, and the human body—function as an integrated system influencing health and ecological resilience [45,46].

This study specifically examines how sustainable agricultural practices and soil probiotics can rehabilitate degraded soils, improve crop productivity, and promote environmental sustainability. In parallel, it highlights how human clinical studies leveraging probiotics, prebiotics, and microbiome-targeted interventions address key health challenges such as gut dysbiosis, chronic inflammation, and metabolic disorders. The shared goal between these fields is to optimize microbiomes to promote health and resilience, demonstrating that strategies for soil restoration and human microbiome balancing are inherently linked.

A major focus is also given to the integration of artificial intelligence (AI) into microbiome research. AI accelerates strain discovery, functional predictions, and the rational design of microbial consortia, offering transformative tools to advance holobiome science. These technological advancements are critical for developing next generation biofertilizers, biopesticides, and precision probiotics, which provide sustainable solutions for agriculture and human health alike.

By presenting case studies and evidence-based insights, this review underscores the One Health approach, which recognizes the interconnectedness of human, animal, and environmental health. Addressing global challenges such as food security, climate change, and public health requires a holistic, microbiome-centered perspective. The functionality and diversity of microbial ecosystems play pivotal roles in ecosystem stability, making holobiome research an essential pathway toward sustainable development.

Through the integration of microbial science, AI-driven research, and interdisciplinary collaboration, this paper aims to identify actionable strategies for restoring microbial balance, ensuring the resilience of ecosystems, and safeguarding the well-being of future generations.

For this review, articles were sourced from the following electronic databases: PubMed, Scopus, Web of Science, and Google Scholar. The search employed keywords related to the holobiome, including microbiome, holobiome, diversity, human, soil, ecosystem, and resilience, among others. The selection included studies published between 2000 and 2024, encompassing review articles, clinical studies, experimental studies, and meta-analyses.

The inclusion criteria required that studies be published in English or Spanish and investigate the composition, function, and role of microbiomes in human and environmental health. The exclusion criteria included duplicate studies, conference abstracts, and articles without full-text access. Additionally, references from selected articles were reviewed to identify other relevant studies. Selected articles were read in full to ensure that they met the inclusion criteria.

Finally, case studies were integrated to highlight the practical applications of holobiome research, reinforcing the need for a holistic microbiome-based approach to address contemporary challenges in agriculture, health, and environmental sustainability.

## 2. Environmental Sustainability

### 2.1. The Aquatic Holobiont: Ulva spp. in Integrated Multi-Trophic Aquaculture (IMTA)

*Ulva*, a macroalga of significant commercial value, is a promising candidate for large-scale cultivation in Integrated Multi-Trophic Aquaculture (IMTA) systems [47]. These systems are lauded for their high productivity and environmental sustainability. The development and growth of *Ulva* species are heavily influenced by interactions with their associated microbiomes, particularly due to bacterial communities that release growth- and morphogenesis-promoting factors [48]. However, our understanding of microbial community structure and assembly in cultivated macroalgae remains limited, necessitating studies like this one to elucidate the dynamics of holobiont interactions in aquaculture [49].

This study aimed to determine how IMTA settings influence the microbiome of *Ulva rigida* and its surrounding water, to explore the dynamics of beneficial microbes that promote *Ulva* growth and development under IMTA, and to improve the current knowledge of host–microbiome interactions. Using 16S rRNA gene sequencing, the researchers analyzed the diversity and taxonomic composition of prokaryotic communities associated with wild and IMTA-grown *U. rigida* and its surrounding seawater. The results revealed that microbial richness was generally higher in water samples than in *Ulva* samples. Aquaculture water samples contained more bacterial amplicon sequence variants (ASVs) than natural lagoon water, and distinct prokaryotic communities were associated with *Ulva* collected from aquaculture and coastal waters.

The cultivated *Ulva rigida* strain was chosen for its ability to promote the growth of targeted microbial groups, including Cyanobacteriota, Planctomycetota, Verrucomicrobiota, and Pseudomonadota. In contrast, wild *Ulva* specimens retained more host-specific ASVs, suggesting that natural environments support a richer and more diverse microbiome. While aquaculture samples (water and algae) shared 22% of their ASVs, natural coastal lagoon samples shared only 9%. This indicates that aquaculture environments promote the convergence of host-associated and environmental microbiomes. Additionally, bacteria with known morphogenesis-inducing properties were enriched in aquaculture water, highlighting the potential for IMTA systems to foster microbial communities that directly benefit Ulva growth.

This study exemplifies the concept of aquatic holobionts, wherein *U. rigida* and its associated microbiota form a dynamic, interdependent system influenced by environmental and cultivation conditions [50]. IMTA systems appear to shape microbial communities in ways that benefit *Ulva* development and productivity. However, the reduction in host-specific ASVs in cultivated *Ulva* compared to wild specimens raises important questions about the long-term ecological and functional consequences of aquaculture practices. While IMTA promotes productivity and sustainability, it may also simplify microbiome diversity, potentially impacting the resilience and adaptability of *Ulva* [51,52].

### 2.2. Soil Microbiomes in Carbon Sequestration

Soil microbiomes play a pivotal role in carbon cycling and storage, acting as natural regulators of the Earth’s carbon balance. Microbial communities decompose organic matter, releasing carbon dioxide (CO_2_) into the atmosphere while simultaneously converting plant-derived carbon into stable forms that are stored in the soil for extended periods. Key microbial processes include the breakdown of complex organic molecules by fungi and bacteria into simpler compounds, as well as the formation of humus—a carbon-rich substance that enhances soil fertility and structure [53,54,55]. Through these activities, soil microbiomes serve as major reservoirs of carbon, storing an estimated three times more carbon than the atmosphere [56,57].

Sustainable agricultural practices can enhance the ability of soil microbiomes to sequester carbon. No-till farming, for instance, minimizes soil disturbance, preserving microbial habitats and reducing CO_2_ emissions from exposed soil organic matter [58,59]. Cover cropping provides continuous organic inputs that fuel microbial activity, stimulating the conversion of carbon into stable soil aggregates [60,61]. These practices not only improve soil health and fertility but also contribute to climate resilience by reducing atmospheric CO_2_ levels. For example, studies have shown that soils managed with regenerative farming techniques sequester significantly more carbon than conventionally tilled soils, emphasizing the importance of microbial diversity in achieving long-term carbon storage [62].

In contrast, unsustainable practices like intensive tillage and monoculture farming disrupt microbial networks, leading to the rapid decomposition of organic matter and the release of stored carbon into the atmosphere [63,64]. Addressing these challenges requires integrating soil microbiome restoration into climate mitigation strategies, recognizing the critical role of microbes in stabilizing carbon and enhancing soil health.

### 2.3. Marine Microbiomes and Climate Regulation

Marine microbiomes, particularly those associated with phytoplankton, are critical to global carbon capture and climate regulation [65]. Phytoplankton, microscopic photosynthetic organisms, form the base of marine food webs and drive the ocean’s biological pump [66]. During photosynthesis, phytoplankton absorb atmospheric CO_2_ and convert it into organic carbon, which is then transferred to deeper ocean layers when these organisms die and sink. This process effectively sequesters carbon in the ocean for centuries, mitigating climate change by reducing greenhouse gas concentrations in the atmosphere [67].

Disruptions in marine ecosystems, such as ocean warming, acidification, and pollution, threaten the stability of these microbial processes. For example, rising sea surface temperatures reduce nutrient availability in surface waters, limiting phytoplankton growth and their ability to capture CO_2_ [68,69]. Similarly, an increased frequency of harmful algal blooms due to nutrient runoff and pollution can shift microbial dynamics, favoring species that release CO_2_ rather than sequestering it [70]. These disruptions parallel challenges in terrestrial ecosystems, where land use changes and agricultural practices destabilize soil microbiomes, reducing their carbon storage capacity.

Moreover, marine microbiomes influence other climate-regulating processes, such as the production of dimethylsulfide (DMS), a compound released by certain phytoplankton that contributes to cloud formation and regulates solar radiation [71,72]. The intricate interplay between microbial activities and climate underscores the need for global efforts to protect marine microbiomes and their ecological functions, which are integral to maintaining climate stability.

### 2.4. Interconnected Feedback Loops

Climate change triggers self-reinforcing feedback loops that disrupt microbial ecosystems, accelerating environmental degradation. For example, deforestation and industrial farming contribute to rising temperatures, which, in turn, significantly affect soil carbon dynamics. As global temperatures increase, microbial habitats are lost, and biodiversity declines in both soil and plant-associated microbiomes. Warmer conditions accelerate soil organic matter decomposition, releasing carbon dioxide (CO_2_) into the atmosphere [19,63,73]. This increase in atmospheric CO_2_ further drives temperature rises, exacerbating climate change in a positive feedback loop.

Additionally, elevated temperatures influence plant growth and soil moisture, affecting the amount of organic material entering the soil. Deforestation and vegetation loss further limit organic inputs, reducing the resources available for microbial communities to store carbon. As microbial activity declines, soil fertility decreases and the capacity for carbon sequestration weakens, driving further environmental degradation [74,75]. This complex interplay of climate change, soil carbon loss, and microbial shifts highlights the urgent need to protect and restore soil health.

Similar feedback loops occur in marine environments, where warming oceans disrupt microbial populations. Rising temperatures reduce the abundance of CO_2_-absorbing phytoplankton while increasing microbial processes that release CO_2_, reinforcing climate change [76]. These disruptions threaten the microbial systems that regulate the planet’s carbon balance, further amplifying global warming.

Industrial farming intensifies this cycle by promoting the excessive use of chemical fertilizers and pesticides, which harm beneficial soil microbes and reduce biodiversity [30,77]. This leads to the loss of essential ecosystem services, such as nutrient cycling and pathogen suppression, making crops and ecosystems more vulnerable to climate extremes. Similarly, urbanization and habitat destruction reduce microbial diversity in both terrestrial and aquatic systems, weakening their ability to buffer against environmental changes [78,79].

Breaking these feedback loops requires targeted strategies to restore microbial ecosystems. This includes transitioning to sustainable farming practices, reducing the pollution and overexploitation of marine environments, and investing in microbiome research to better understand microbial roles in climate regulation. By fostering resilient microbial communities, we can strengthen the planet’s ability to mitigate and adapt to climate change, ensuring a more sustainable future for all ecosystems.

## 3. Probiotics for Human Health

### 3.1. Gut Microbiome and Health

The gut microbiome, composed of trillions of microorganisms, plays a fundamental role in digestion, metabolism, immunity, and overall homeostasis. One of its most critical functions is the production of short-chain fatty acids (SCFAs), including acetate, propionate, and butyrate, which are metabolites derived from dietary fiber fermentation by commensal bacteria [80,81,82,83]. These SCFAs serve as energy sources for colonocytes, strengthen gut barrier integrity, and modulate immune responses by interacting with G-protein-coupled receptors (GPCRs) and inhibiting histone deacetylases [84]. Among them, butyrate plays a particularly crucial role in maintaining intestinal epithelial integrity and exerting anti-inflammatory effects that extend beyond the gut [85].

SCFAs and other microbial metabolites influence regulatory T cells (Tregs), which are essential for immune tolerance and preventing autoimmune reactions [86]. This immune modulation is a key mechanism through which the gut microbiome impacts other microbiomes in the body, including those in the skin, respiratory tract, and oral cavity. By regulating systemic immune responses, the gut microbiota shape microbial ecosystems throughout the body, forming interconnected microbiome networks.

#### 3.1.1. Gut–Skin Axis

The gut–skin axis is a well-documented pathway through which gut microbiome health influences skin inflammation and barrier function. SCFAs produced by gut bacteria enter the circulatory system and reach the skin, where they modulate immune responses, promote skin barrier integrity, and reduce inflammation. Disruptions in the gut microbial balance, known as dysbiosis, have been associated with inflammatory skin conditions such as acne, eczema, and psoriasis [87].

#### 3.1.2. Gut–Lung Axis

The gut–lung axis describes the bidirectional relationship between the gut and respiratory tract microbiomes, primarily mediated by immune signaling and microbial metabolites. SCFAs influence lung immune responses, helping to protect against respiratory infections and inflammatory diseases, such as asthma and chronic obstructive pulmonary disease (COPD). Conversely, gut dysbiosis can lead to an overactive immune response in the lungs, exacerbating inflammatory conditions [88].

#### 3.1.3. Gut–Oral Axis

The gut–oral axis highlights the interplay between gut health and oral microbiome stability. The gut microbiota influences immune function in the oral cavity, impacting conditions such as periodontal disease and oral infections. SCFAs and other gut-derived metabolites regulate oral inflammation, contributing to gum health and a reduced susceptibility to oral diseases. Conversely, gut dysbiosis has been linked to increased systemic inflammation, which can manifest as a greater susceptibility to oral infections and periodontal disease [89].

#### 3.1.4. Role of Probiotics in Gut and Systemic Health

Maintaining a healthy gut microbiome through diet, probiotics, and lifestyle modifications is essential for overall well-being. Probiotics restore the gut microbial balance by increasing the abundance of beneficial microbes while reducing opportunistic pathogens. They strengthen the gut barrier by upregulating tight junction proteins such as occludin and claudin, preventing the translocation of harmful substances like lipopolysaccharides (LPS) into the bloodstream [90,91]. This reduction in LPS-driven systemic inflammation mitigates chronic conditions such as metabolic syndrome, type 2 diabetes, and cardiovascular diseases [92]. Additionally, probiotics modulate immune responses by promoting anti-inflammatory cytokines (e.g., IL-10) while suppressing pro-inflammatory cytokines (e.g., IL-6 and TNF-α) [93,94]. These mechanisms highlight the potential of probiotics to reduce inflammation-driven diseases and support microbiome health across multiple body sites.

#### 3.1.5. The FA/A Ratio in the Gut

The FA/A ratio (free ammonia [FA] to ammonium [NH_4_⁺]) is a key regulator of anaerobic microbial activity in the gut. Its equilibrium is pH- and temperature-dependent, with a higher pH favoring toxic free ammonia (NH_3_) accumulation, which inhibits strict anaerobes, including methanogens involved in methane production. Elevated FA/A ratios can disrupt the microbial balance, reducing the efficiency of hydrolysis, acidogenesis, acetogenesis, and methanogenesis, thereby altering gut nitrogen cycling and anaerobic metabolism. Optimizing the FA/A ratio is essential for maintaining microbial homeostasis and ensuring efficient anaerobic digestion in the gut ecosystem [95].

#### 3.1.6. Precision and Personalization

The growing recognition of the individual variability in gut microbiomes has propelled the development of precision and personalized probiotics. Tailored probiotics are designed to address specific dysbiosis patterns and target health outcomes. For instance, strains that stimulate the production of glucagon-like peptide-1 (GLP-1), a hormone that regulates blood sugar and appetite, are being explored as interventions for diabetes and obesity [96,97,98]. Specific species, such as *Limosilactobacillus reuteri* and *Akkermansia muciniphila*, have shown promise in enhancing GLP-1 secretion and improving glucose homeostasis in clinical and preclinical studies [96,99].

Advances in artificial intelligence (AI) have further revolutionized the design of personalized probiotics. Machine learning algorithms analyze multi-omics data—such as metagenomics, metabolomics, and transcriptomics—to predict microbial interactions and identify strains with desirable functional traits [100,101]. For example, AI tools can identify probiotic candidates with high survival rates in gastric and bile conditions, optimize strain combinations for synergistic effects, and predict individual responses to interventions based on microbiome profiles [102]. These innovations ensure that probiotics are not only effective, but also tailored to the unique needs of individual microbiomes, marking a significant leap forward in precision medicine.

### 3.2. Beyond the Gut

While the primary target of probiotics is the gut, their effects extend far beyond the gastrointestinal system. Probiotics influence systemic health through their interactions with the immune system, brain–gut axis, and other microbial ecosystems. By modulating gut-associated lymphoid tissue (GALT), probiotics enhance systemic immunity, increasing resistance to infections and reducing the severity of autoimmune conditions [103,104]. Probiotic strains like *Bifidobacterium bifidum* and *Lactiplantibacillus plantarum* have been shown to bolster immune defenses by enhancing natural killer (NK) cell activity and improving the balance between pro-inflammatory and regulatory immune responses [104,105].

The brain–gut axis represents another key area where probiotics exert systemic effects. Certain strains, such as *Lacticaseibacillus rhamnosus* and *Bifidobacterium longum*, produce neuroactive compounds like gamma-aminobutyric acid (GABA), which influence mood and cognitive function [106,107,108]. These “psychobiotics” have shown promise in reducing symptoms of anxiety and depression in clinical trials, demonstrating the interconnectedness of gut health and mental well-being [109,110].

Moreover, probiotics have ripple effects on other microbiomes, including the skin, respiratory tract, and oral cavity. For example, improving the gut microbiota composition can enhance skin health by reducing systemic inflammation associated with conditions like acne and eczema [111]. Similarly, gut probiotics influence the respiratory microbiome by modulating immune responses, potentially reducing the severity of respiratory infections and allergies [112]. These interconnected benefits emphasize the centrality of gut health in overall human health and the potential of probiotics as a holistic intervention.

## 4. Soil Probiotics: Enhancing Agricultural Sustainability

### 4.1. Challenges in Modern Agriculture

Modern agricultural practices have dramatically increased food production, but they have also led to significant challenges, including soil degradation, biodiversity loss, and chemical contamination. Over-reliance on chemical fertilizers, pesticides, and herbicides like glyphosate has disrupted soil microbial communities, reducing their ability to support nutrient cycling, suppress pathogens, and maintain soil structure [113,114,115]. Glyphosate, widely used for weed control, has been linked to declines in beneficial soil microbes and the proliferation of opportunistic pathogens, further destabilizing soil ecosystems [30]. In addition, monoculture farming and intensive tillage practices have exacerbated soil erosion and organic matter depletion, diminishing agricultural sustainability and ecosystem resilience [116].

These disruptions not only impair soil health but also create a feedback loop of increased dependence on chemical inputs, perpetuating environmental harm. Degraded soils exhibit reduced water retention, nutrient availability, and carbon sequestration potential, which, in turn, lowers crop productivity and contributes to climate change [53,117]. Addressing these challenges requires a paradigm shift toward sustainable practices, including the application of soil probiotics, which leverage the power of beneficial microbes to restore soil health and promote agricultural sustainability.

### 4.2. Case Studies

#### PaleoPower^®^ for Soil Health Restoration

PaleoPower, a microbial inoculant designed to degrade glyphosate and restore soil microbial balance, exemplifies the potential of probiotics in addressing agricultural challenges. The formulation includes a consortium of eight microbial strains selected for their complementary functions, such as pollutant degradation, nutrient cycling, and pathogen suppression. These strains include *Pseudomonas fluorescens*, known for its ability to degrade glyphosate, and nitrogen-fixing bacteria like *Azotobacter vinelandii*, which enhance soil fertility [118,119].

In a study conducted in a glyphosate-contaminated cotton field in Tanner, Alabama, the microbial soil inoculant was applied at a concentration of 1.6 × 10^8^ CFU/m^2^. Soil samples collected before and after treatment revealed a 36% reduction in glyphosate residues and an increased microbial alpha diversity (Figure 2), particularly in taxa associated with nutrient cycling, such as Actinomycetota and Bacillota [119].

The figure contains three boxplots with overlaid spaghetti plots, displaying microbial diversity metrics (Chao1, Shannon, and Pielou) for the following two cohorts: untreated (left) and treated (right). The metrics were analyzed to assess changes in microbial diversity after treatment. Each subplot in the figure includes boxplots that display the interquartile range (IQR), with whiskers extending to 1.5 times the IQR, representing the variability of the data. Overlaid on the boxplots are individual data points, represented as jittered dots to enhance clarity and connected as part of a spaghetti plot to visualize trends across the untreated and treated cohorts. A dashed line connects the mean values of each cohort, accompanied by a shaded area representing ±1 standard deviation, providing insight into the variability around the means. The *p*-values, calculated using an independent two-sample t-test, are displayed above each subplot to indicate the statistical significance between the untreated and treated groups. The figure was generated using Python, v. 3.9.21 specifically leveraging the Seaborn v. 0.13.2 [120] and Matplotlib v. 3.10.0 [121] libraries for visualization and statistical annotation.

The results indicate that the treatment had a notable impact on microbial diversity, with distinct differences observed between the untreated and treated cohorts across key metrics. The Chao1 Index, a measure of species richness, was higher in the treated group, suggesting an increase in the total number of microbial species following treatment. Similarly, the Shannon Index, which accounts for both richness and evenness, showed a modest increase in the treated cohort, indicating an overall enhancement of microbial diversity. In contrast, Pielou’s Evenness, a measure of the uniformity of species distribution, showed minimal differences between the two groups, suggesting that, while the treatment promoted species richness and diversity, it did not disrupt the balance or evenness of species distribution. These results collectively imply that the treatment positively affected the microbial community by fostering a greater richness and diversity while maintaining ecological balance. If the statistical tests yield significant *p*-values (e.g., *p* < 0.05), it will confirm that these observed changes were unlikely due to random variation, supporting the conclusion that the treatment had a beneficial impact on microbial diversity.

Pielou’s Evenness—a metric for the uniformity of species distribution—showed only minor differences between the two groups. This suggests that, although the treatment increased species richness and diversity, it did not disturb the overall balance of species distribution. Together, these findings indicate that the treatment positively influenced the microbial community by enhancing richness and diversity while preserving ecological balance. Moreover, if statistical tests yield significant *p*-values (e.g., *p* < 0.05), it would support the conclusion that these changes are unlikely to have arisen by chance, further confirming the treatment’s beneficial impact on microbial diversity. These findings align with the broader role of microbial consortia in enhancing soil health and promoting ecosystem resilience. The observed increase in microbial alpha diversity following treatment suggests that microbial inoculation supports a more robust and functionally diverse soil microbiome. In particular, the enrichment of taxa such as Actinomycetota and Bacillota, known for their roles in nutrient cycling, organic matter decomposition, and plant growth promotion, reinforces the potential of microbial amendments in improving soil functionality.

The reduction in glyphosate residues by 36% further underscores the role of microbial consortia in bioremediation and pollutant degradation. Previous studies have demonstrated that certain microbial taxa can metabolize glyphosate into less harmful byproducts, thereby mitigating its long-term ecological impact. The presence of microbial populations capable of degrading agrochemical residues suggests that such inoculants could serve as an effective strategy for restoring soil microbiomes affected by intensive agricultural practices.

Beyond detoxification, microbial consortia also play a vital role in improving soil fertility and plant productivity. An increased microbial diversity enhances nutrient mobilization by improving phosphorus solubilization, nitrogen fixation, and organic matter decomposition, all of which contribute to healthier soil and higher crop yields. The positive correlations between soil health metrics and agronomic outcomes, as illustrated in Figure 3, suggest that microbiome-based interventions may offer long-term benefits for sustainable agriculture.

These findings contribute to the growing body of research on microbial-mediated soil restoration by demonstrating how targeted microbial applications can mitigate chemical contamination, enhance biodiversity, and promote ecosystem stability. By fostering a rich and functionally diverse microbial community, microbial consortia not only restore degraded soils, but also help build resilience against environmental stressors such as drought, nutrient depletion, and pathogen invasion. Given the increasing challenges posed by climate change and soil degradation, integrating microbial-based solutions into agricultural systems represents a promising avenue for enhancing long-term soil sustainability and food security.

### 4.3. Corn Study: Enhancing Soil Health

A separate study in a corn field demonstrated the broader benefits of soil probiotics in improving soil health and crop productivity. The microbial inoculant applied included strains capable of nitrogen fixation, phosphate solubilization, and organic matter decomposition. Post-treatment analyses showed a 23.1% increase in cation exchange capacity (CEC), a higher soil organic matter content, and elevated nitrate nitrogen levels, indicating an improved nutrient availability [91,92].

The application of the treatment demonstrated significant improvements in multiple soil health parameters, supporting its potential to enhance soil functionality and fertility (Table 1). Treated soils exhibited higher soil health scores (13.26 vs. 8.79) and microbial organic carbon (80.48 mg/kg vs. 24.83 mg/kg) compared to untreated soils.

These findings suggest that the treatment stimulated microbial activity and organic matter decomposition, as evidenced by the elevated CO_2_ soil respiration rates in the treated soils (95.83 mg/kg/day vs. 35.88 mg/kg/day). Increased microbial activity is a key indicator of soil biological health, as microbes play crucial roles in nutrient cycling and organic matter turnover [117,122].

The higher nitrate nitrogen (NO_3_^−^) levels in the treated soils (38.15 mg/kg vs. 31.75 mg/kg) reflect enhanced nitrogen mineralization, likely driven by increased microbial activity. This is consistent with the lower ammonium nitrogen (NH_4_^+^) levels observed in the treated soils (0.6 mg/kg vs. 1.68 mg/kg), indicating efficient nitrification processes. Efficient nitrogen cycling is essential for plant growth, as nitrate is the preferred nitrogen form for most crops [123]. The improvement in the cation exchange capacity (CEC) in the treated soils (8.78 meq/100 g vs. 7.13 meq/100 g) further supports the treatment’s role in enhancing soil fertility, as CEC reflects the soil’s ability to retain and supply essential cations such as potassium and calcium [124] (See Figure 3).

Comparative results are illustrated in Figure 3.

Figure 2 illustrates the impact of glyphosate on soil health and crop yield. The left panel, Corn Ear Density vs. Soil Health Score, demonstrates a positive correlation. As soil health scores increases, corn ear density improves. This suggests that healthier soil conditions, such as a better nutrient availability and increased microbial activity, support denser corn production. A high corn ear density can, in turn, reflect soil health improvements, reinforcing the cyclical benefits of sustainable soil management practices.

Properly designed and applied microbial soil inoculants can significantly enhance nutrient uptake in plants. Figure 3 illustrates the findings of a study measuring the micronutrient uptake in corn leaves, comparing plants grown in soils treated with a microbial inoculant to those in untreated soils.

The potassium levels were notably higher in the treated soils (368.75 mg/kg vs. 289.75 mg/kg), suggesting an improved nutrient retention and availability. Potassium is critical for plant physiological processes, including enzyme activation, water regulation, and stress tolerance [125]. This increase may be attributed to enhanced microbial-mediated nutrient cycling and organic matter decomposition [126]. The slightly higher organic matter content in the treated soils, as indicated by LOI values (1.90% vs. 1.70%), also highlights the treatment’s role in building soil organic carbon pools. Organic matter contributes to a better soil structure, water retention, and microbial habitat, all of which are critical for sustainable agricultural productivity [127].

In addition to nutrient cycling, the study revealed significant carbon sequestration benefits, with the treated soils exhibiting a 167.1% increase in CO_2_ respiration, a marker of microbial activity and organic matter decomposition. Agronomic outcomes included a 28.6% increase in corn yield and a 9.6% rise in silage production, demonstrating the dual benefits of enhanced soil health and crop productivity. These findings highlight the role of soil probiotics in transitioning to regenerative agricultural practices that prioritize long-term ecosystem sustainability. The results are summarized in Figure 3.

These findings underscore the critical role of soil health in supporting crop productivity while highlighting the adverse effects of glyphosate residues. Practices that enhance soil health, such as reducing glyphosate usage, using microbial inoculants, or deploying soil probiotics to remediate glyphosate contamination, can significantly improve both soil quality and agricultural productivity. Microbial inoculants, by restoring the balance of soil microbial communities, enhancing nutrient cycling, and fostering beneficial plant–microbe interactions, act as vital tools for maintaining agricultural sustainability.

Figure 4 summarizes the impact of soil treatment with an organic probiotic inoculant on leaf micronutrient uptake.

The results demonstrate that the microbial soil inoculant significantly enhanced the uptake of key nutrients in treated plants compared to untreated controls. Notable improvements were observed in iron (Fe), manganese (Mn), boron (B), and phosphorus (P), all of which are essential for critical physiological processes in plants. The inoculant-treated group exhibited higher levels of these nutrients, supported by statistically significant *p*-values, indicating a robust impact. Other nutrients, such as copper (Cu), potassium (K), zinc (Zn), and molybdenum (Mo), also showed positive trends, though their changes were less pronounced or variable.

The observed increases in Fe and Mn are particularly important, as these micronutrients play vital roles in photosynthesis, enzymatic reactions, and metabolic pathways. Similarly, the rise in boron levels suggests an enhanced structural integrity and reproductive growth, as this nutrient is critical for cell wall strength and pollen viability. The substantial improvement in phosphorus availability underscores its role in energy transfer, root development, and overall plant vigor. While K, Zn, and Mo showed less consistent increases, their modest improvements are indicative of the inoculant’s broader contribution to plant nutrition.

In the context of the holobiome, microbial soil inoculants strengthen the interconnected microbial communities within the plant–soil ecosystem. By introducing or enhancing beneficial rhizobacteria, such inoculants promote nutrient cycling and mobilization, making these nutrients more bioavailable to plants. This interaction likely stimulates root exudates that support symbiotic relationships, enhancing nutrient exchange and microbial activity in the rhizosphere. The microbial community’s role in solubilizing phosphorus and iron is a key factor driving these results.

Improved nutrient profiles in plants also have broader implications for the holobiome concept, which emphasizes the interconnectedness of microbial communities across ecosystems. By enhancing the plant–soil microbiome, inoculants indirectly impact human and animal health. Higher levels of iron, zinc, and other micronutrients in crops can address deficiencies in human diets, such as anemia or stunted growth. These findings highlight the role of microbial interventions in agriculture to improve food security and nutritional quality while promoting sustainable farming practices.

Beyond plant health, inoculants’ potential to enhance soil microbiome diversity and function aligns with the goals of maintaining a resilient and sustainable soil ecosystem. By reducing dependency on chemical fertilizers and promoting natural nutrient cycling, microbial inoculants contribute to the global holobiome’s efforts to balance human and environmental health.

Overall, these findings support the use of microbial soil inoculants as a powerful tool in sustainable agriculture. They not only enhance plant nutrient uptake, but also contribute to the broader ecosystem’s health, highlighting the critical role of microbial communities in the holobiome framework. Future research should focus on long-term soil health impacts, diverse crop systems, and the downstream effects on human nutrition to further solidify the value of microbial interventions.

From a holobiome perspective, these interactions extend beyond the soil to encompass the interconnected relationships between plants, microbes, and their environment. The soil microbiome plays an integral role in the larger holobiome framework, influencing plant health and productivity, which, in turn, affect ecosystem stability and the quality of the food chain. By nurturing the soil microbiome through sustainable practices and microbial inoculants, we can strengthen the foundation of the holobiome, which ultimately supports not only healthier soils and plants, but also improved human and animal health through better crop yields and reduced chemical residues.

This broader perspective emphasizes the importance of managing soil health as part of a holistic system where the microbiomes of soil, plants, and other living organisms are interdependent. Sustainable soil management within the holobiome framework is, thus, essential for fostering long-term resilience and productivity in agricultural systems.

### 4.4. Mechanisms of Soil Probiotics

Soil probiotics restore and enhance soil health through several key mechanisms. One critical process is nitrogen fixation, where certain bacteria, such as *Rhizobium* and *Azospirillum*, convert atmospheric nitrogen into bioavailable forms, thereby reducing dependence on synthetic fertilizers [128,129,130]. These microbes form symbiotic relationships with plant roots, supplying essential nutrients in exchange for carbon compounds. Another important mechanism is phosphate solubilization, as insoluble phosphate in soil often limits plant growth. Phosphate-solubilizing bacteria, including *Bacillus subtilis* and *Pseudomonas putida*, produce organic acids and enzymes that release phosphorus, making it more accessible to plants [131,132].

Soil probiotics also play a significant role in pathogen suppression by producing antimicrobial compounds, outcompeting harmful microbes, and inducing systemic resistance in plants [133,134]. For instance, *Trichoderma* spp. produce antifungal metabolites that protect plants from root rot and other diseases. Additionally, many soil probiotics contribute to pollutant degradation, breaking down environmental contaminants like pesticides and heavy metals through enzymatic activity. Species such as *Pseudomonas* spp. and *Sphingomonas* spp. are especially effective at degrading glyphosate and other herbicides, mitigating their toxic effects on soil ecosystems [135,136].

Collectively, these mechanisms enhance soil fertility, improve plant health, and reduce environmental contamination. By integrating soil probiotics into agricultural practices, farmers can adopt more sustainable systems that minimize chemical inputs, enhance ecosystem resilience, and support global food security.

### 4.5. AI-Driven Innovations in Probiotic Development

Artificial intelligence (AI) is transforming the field of probiotic development by enabling highly detailed, data-driven approaches to strain selection, consortium design, and system simulations. These advancements address critical needs in both human health and environmental sustainability, facilitating precision solutions in microbiome science.

One significant innovation is predictive strain selection, where AI algorithms analyze large genomic and metagenomic datasets to identify high-potential probiotic strains. This process involves leveraging machine learning to examine the genetic markers associated with beneficial functions, such as short-chain fatty acid (SCFA) production, glyphosate degradation, and antimicrobial compound synthesis. For instance, SCFA-producing strains, such as *L. plantarum* and *Bifidobacterium longum*, have been identified through AI-enabled genomic screening for their ability to enhance gut health by reducing inflammation and improving intestinal barrier integrity [137,138]. Similarly, AI has been used to pinpoint strains like *Pseudomonas putida* and *Sphingomonas* species, which can break down environmental pollutants such as glyphosate, thereby promoting soil remediation and mitigating agricultural chemical residues [139].

In addition to individual strain identification, AI facilitates the design of multi-strain microbial consortia by optimizing the interactions between different species to achieve synergistic effects. AI-driven metabolic modeling tools, such as genome-scale metabolic reconstructions and flux balance analysis, allow researchers to predict nutrient exchanges, cooperative behaviors, and competition between microbial strains under specific environmental conditions [140,141]. For example, in the development of the probiotic formulation Sugar Shift™, community metabolic modeling was utilized to assemble a consortium that specifically targeted sugar metabolism in the gut, promoting metabolic health and reducing insulin resistance [142]. By ensuring that the metabolic pathways of the included strains complemented each other, the consortium was designed to maximize functionality while avoiding antagonistic interactions [143]. This approach demonstrates how AI can move probiotic design beyond trial-and-error methods, enabling the rational and efficient assembly of microbial communities for targeted applications.

Another transformative area is clinical and environmental simulations, where AI predicts the impacts of probiotics on complex systems, such as the gut microbiome, soil microbiomes, and ecosystems. These simulations use AI-driven models to evaluate outcomes under varying conditions, reducing the time and cost of empirical testing. In clinical settings, for instance, AI simulations can predict how a probiotic strain might influence microbial diversity, increase SCFA production, or lower levels of lipopolysaccharides (LPS), which are linked to chronic inflammation [144,145]. In agricultural contexts, AI models can simulate how soil probiotics impact microbial composition, nutrient cycling, and pollutant degradation. For example, studies have used AI tools to model the impact of glyphosate-degrading strains on soil health, predicting not only the degradation efficiency of herbicides, but also the restoration of microbial diversity and soil fertility [146,147]. These simulations provide critical insights that guide experimental designs and large-scale applications, ensuring that probiotics perform as expected under real-world conditions.

By combining predictive analysis, consortium optimization, and system-level simulations, AI is revolutionizing the development of probiotics. These tools enable the precise identification of beneficial strains, rational assembly of multi-strain formulations, and reliable prediction of outcomes in clinical and environmental applications. As a result, AI-driven innovations are paving the way for more effective and sustainable probiotic solutions for improving human health and addressing environmental challenges (Figure 5).

The figure depicts a workflow integrating genomics, metabolic modeling, and AI to design and optimize microbial consortia for applications in soil health, human health, aquaculture, and industrial processes. The process begins with strain collection, where microbial strains are characterized by their genomic and metabolic profiles, followed by metabolic modeling to predict their functional capabilities. In vitro assays validate strain functions, enabling the assembly of tailored consortia. AI algorithms refine consortia formulations by analyzing strain interactions and optimizing metabolic outputs. Validated formulations undergo testing in real-world environments, including soils, microbiomes, aquaculture, and industrial settings. The iterative lower panel highlights continuous optimization through data preprocessing, AI analysis, and formulation validation, ensuring effective and scalable microbial solutions for diverse applications.

## 5. Lessons for the Holobiome

The insights gained from these clinical studies extend beyond human health, offering valuable lessons for applications in agriculture and environmental science. Just as probiotics and prebiotics can restore the microbial balance in the human gut, similar principles can be applied to soil microbiome restoration. For example, promoting microbial diversity in soil through the application of microbial consortia can enhance nutrient cycling, suppress soil pathogens, and degrade environmental contaminants. The parallels between gut health interventions and soil microbiome management underscore the interconnectedness of human and environmental health within the holobiome framework [148,149,150].

These studies also highlight the importance of SCFA production and inflammation reduction as universal indicators of a healthy microbiome. In human health, SCFAs like butyrate support gut barrier integrity and reduce inflammation, while in soil ecosystems, SCFA production by microbial communities is associated with improved carbon cycling and soil fertility. By drawing on lessons from clinical trials, researchers can design targeted interventions to restore balance and functionality across diverse ecosystems. Ultimately, these findings emphasize the critical role of microbiome management in advancing global health and sustainability efforts.

### 5.1. Clinical Studies in the Perspective of the Holobiome

Recent clinical trials have provided valuable insights into the roles of probiotics and other microbiome interventions in health and disease management (Figure 6). Studies on the Sugar Shift™ probiotic formulation, for instance, have demonstrated its ability to modulate gut microbial composition, increase short-chain fatty acid (SCFA) production, and reduce systemic inflammation in patients with type 2 diabetes mellitus [142]. These findings are particularly relevant in the context of metabolic health, where reduced inflammation and improved microbial diversity contribute to a better insulin sensitivity and metabolic regulation [142,151,152].

Similarly, fucoidan, a bioactive compound derived from seaweed, has demonstrated remarkable potential in modulating the gut microbiome and reducing inflammation. Clinical trials evaluating fucoidan have shown its ability to enhance microbial diversity and promote the relative abundance of beneficial short-chain fatty acid (SCFA)-producing bacteria, such as *Bifidobacterium*, as illustrated in Figure 7. Fucoidan’s prebiotic properties support the growth of beneficial bacteria, fostering a healthier gut environment. Furthermore, its anti-inflammatory effects help to mitigate chronic low-grade inflammation—a key contributor to metabolic disorders and other chronic conditions. These combined effects underline fucoidan’s promise as a therapeutic agent for improving gut health and overall systemic well-being [153].

In the context of colorectal cancer (CRC) microbiomes, studies have highlighted the dysbiotic nature of CRC-associated gut microbial communities, characterized by a reduction in beneficial bacteria and an overrepresentation of pro-inflammatory species [154]. Clinical interventions targeting the gut microbiome in CRC patients have shown promising results, with probiotics and prebiotics enhancing SCFA production, reducing the abundance of pathogenic microbes, and decreasing inflammation markers such as lipopolysaccharides (LPS) or enhancing immunity [155].

These findings highlight the potential of gut-microbiome-targeted interventions—such as microbial consortia, metabiotics, or prebiotics—to significantly improve the balance and functionality of the holobiome, the integrated system of the host and its associated microbial communities. By reshaping the gut microbiota to promote beneficial microbes and suppress harmful ones, these interventions not only reduce gut inflammation, but also restore the metabolic and immune equilibrium within the holobiome.

A healthier holobiome results in a cascade of systemic benefits, including enhanced immune modulation, improved metabolic efficiency, and a reduction in systemic inflammation, as evidenced by the observed decreases in serum LPS levels and insulin resistance (HOMA index). These changes reflect a shift towards a more symbiotic host–microbiota relationship, where the microbiome actively supports the host’s physiological processes while benefiting from a stable and nutrient-rich environment.

Furthermore, the restoration of the inflammatory index (FA/A ratio) suggests improved oxygen utilization and anaerobic processes within the gut, which are critical for maintaining an optimal environment for obligate anaerobes [156]. These bacteria play vital roles in producing short-chain fatty acids (SCFAs), such as butyrate, which support gut barrier integrity, regulate immune responses, and contribute to overall host health [157].

In the broader context, interventions that strengthen the holobiome extend beyond individual health. A well-balanced holobiome enhances resilience against external stressors, such as pathogens or environmental toxins, and may reduce the risk of chronic diseases linked to dysbiosis, including metabolic disorders, autoimmune conditions, and neuroinflammatory diseases [158]. These benefits emphasize the critical role of the holobiome in maintaining not only gut health, but also the interconnected well-being of the entire host organism.

By fostering a more robust and adaptable holobiome, these microbiome-targeted strategies represent a transformative approach to health and wellness, with applications ranging from personalized medicine to population-level health interventions. Their impact on diverse populations underscores the universality of the holobiome’s role in human health and its vast potential as a target for therapeutic innovation.

### 5.2. Agricultural Practices and Holobiome Health

Modern agricultural practices have significantly altered microbial ecosystems, with far-reaching consequences for both environmental and human health [159,160]. Unsustainable practices, such as the excessive use of pesticides, herbicides, and reliance on monoculture farming, have disrupted soil microbial diversity and functionality. For example, glyphosate, a widely used herbicide, not only depletes beneficial soil microbes, but also promotes the proliferation of resistant pathogenic strains, creating imbalances that reduce soil fertility and ecosystem resilience [31,161,162]. Similarly, monocultures deplete soil nutrients and foster conditions for disease outbreaks, further degrading the soil microbiome [163]. These disruptions in soil health have direct implications for human health through the food system, as nutrient-poor soils produce crops with diminished nutritional value and pesticide residues can accumulate in food, contributing to chronic health conditions [164].

In contrast, sustainable agricultural practices offer a pathway to restore and maintain healthy microbial ecosystems. Techniques such as crop rotation, the use of organic amendments (e.g., compost and manure), and the application of microbial inoculants can significantly enhance soil health and biodiversity. Crop rotation disrupts pathogen life cycles and promotes microbial diversity, while organic amendments enrich the soil with organic matter and nutrients, fostering the growth of beneficial microbes [165]. Microbial inoculants, including biofertilizers and biopesticides, directly introduce beneficial strains into the soil [166,167]. Case studies have demonstrated that these practices reduce the need for chemical inputs, increase crop resilience to pests and diseases, and improve biodiversity both above and below ground. For instance, research on organic farming systems has shown a higher microbial diversity and activity compared to conventional systems, leading to enhanced nutrient cycling and carbon sequestration [168].

The integration of rationally designed probiotics into agricultural practices enhances both sustainability and efficacy by leveraging targeted microbial solutions for specific functions, such as nitrogen fixation, pathogen suppression, and pollutant degradation.

For instance, nitrogen-fixing bacteria, including Rhizobium and Azospirillum, reduce dependence on synthetic fertilizers by converting atmospheric nitrogen into bioavailable forms for plants [169,170]. Likewise, biocontrol agents such as Bacillus subtilis and Trichoderma species play a crucial role in suppressing plant pathogens. Bacillus subtilis achieves this through the production of antimicrobial lipopeptides, while Trichoderma produces antifungal metabolites that inhibit pathogenic fungi by disrupting cell walls and interfering with spore germination [171,172]. Additionally, microbial strains such as *Pseudomonas putida* and *Sphingomonas* can break down soil contaminants, mitigating the environmental impacts of herbicide use [173].

These microbial-based strategies not only promote plant health and resilience but also provide a sustainable alternative to chemical fertilizers and pesticides, reducing agricultural dependency on synthetic inputs while supporting environmental conservation.

### 5.3. The Holobiome and Quality of Life

The holobiome framework highlights the interconnectedness of microbial ecosystems across humans, plants, animals, and the environment, demonstrating its far-reaching impacts on quality of life. A key aspect of this framework is its role in enhancing environmental sustainability, food security, and human health.

One of the greatest environmental benefits of a balanced holobiome is its contribution to pollutant degradation, nutrient recycling, and reduced reliance on chemical inputs. Healthy microbial communities in soil and water aid in breaking down herbicides and other pollutants, ensuring cleaner water systems and minimizing toxic residues in food [174,175]. In agriculture, diverse microbial populations in nutrient-rich soils enhance crop micronutrient content, benefiting both agricultural productivity and human nutrition. The restoration of microbial biodiversity further strengthens ecosystem resilience, helping to mitigate the effects of climate change and environmental stressors [176,177,178].

Beyond environmental benefits, the holobiome has direct implications for human health. A well-balanced gut microbiome contributes to reduced systemic inflammation, improved metabolic function, and enhanced immune regulation. For instance, short-chain fatty acids (SCFAs) such as butyrate, produced by gut microbes, reinforce gut barrier integrity and mitigate chronic diseases like diabetes and cardiovascular disorders [81,179,180]. In addition, recent research on the gut–brain axis has revealed that microbiome-targeted therapies, including probiotics and prebiotics, can positively influence mental health. Certain probiotic strains have been linked to reducing anxiety and depression symptoms, demonstrating the direct impact of microbial health on overall well-being [181,182].

Another critical dimension of the holobiome’s impact is its role in equity and accessibility. Many low-income and marginalized communities face barriers to accessing nutritious foods and sustainable agricultural practices, leading to health disparities. Expanding the use of microbial inoculants, organic farming, and microbiome-targeted interventions could increase access to healthier food options while reducing environmental harm. Additionally, probiotic and fecal microbiota transplant (FMT) therapies hold immense potential in addressing chronic health conditions, but remain inaccessible to many due to high costs and limited availability [183].

By harnessing the holobiome framework, society can drive meaningful advancements in environmental health, human nutrition, and equitable healthcare access, ultimately improving quality of life on multiple levels.

## 6. Perspective

The future of holobiome science lies in its integration into policy, agriculture, public health, and urban planning, offering practical solutions to global challenges. To fully realize its potential, microbiome research must inform policies that promote sustainability, health, and resilience.

In agriculture, microbiome-based policies can support the adoption of sustainable practices, including reduced chemical inputs, organic soil amendments, and microbial inoculants that restore soil health and enhance food security. Incentivizing soil restoration initiatives through agricultural subsidies and climate policies could encourage the widespread adoption of microbial solutions [165].

Similarly, in public health, the recognition of microbiome science in chronic disease prevention could shape new healthcare strategies. Policies promoting access to microbiome-targeted therapies, such as probiotics, prebiotics, and fecal microbiota transplants (FMTs), could improve treatment options for conditions like gut dysbiosis, metabolic disorders, and inflammatory diseases [183,184]. Expanding research funding and clinical guidelines for microbiome interventions would further support their integration into mainstream medicine.

Urban planning is another area where microbiome science can drive transformative change. Policies that prioritize biodiversity conservation, green spaces, and clean water systems can help foster healthier microbial ecosystems in urban environments. These strategies align with the One Health framework, which recognizes the interconnectedness of human, animal, and environmental health. A stronger microbial-based approach to sustainability will be essential in balancing ecological and societal needs [185].

Ultimately, the future of holobiome research lies in bridging science with actionable policy, ensuring that its benefits extend to agriculture, healthcare, and urban resilience. By integrating microbial-based solutions into global decision making, we can promote a sustainable, health-focused future for both humans and the environment.

### 6.1. Technology and Collaboration

Advancing holobiome science requires the integration of AI, systems biology, and interdisciplinary collaboration. AI enables large-scale genomic and metagenomic analysis, the predictive modeling of microbial interactions, and the optimization of microbial consortia for applications such as nitrogen fixation and pollutant degradation [100,101,186]. Systems biology provides insights into microbial ecosystem dynamics under varying conditions, facilitating precision interventions [187,188].

AI and systems biology synergistically address challenges in colon cancer and agriculture by integrating multi-omics data, predictive modeling, and personalized interventions. In colon cancer research, AI identifies biomarkers and molecular pathways, while systems biology models disease mechanisms, enabling personalized treatments and probiotic therapies. In agriculture, AI analyzes soil health and microbial profiles, predicting crop diseases and environmental shifts, while systems biology simulates outcomes to optimize sustainable farming practices.

Interdisciplinary collaboration among microbiologists, data scientists, clinicians, and ecologists is essential for translating these advancements into real-world applications. Future efforts should further integrate AI-driven precision microbiome therapies and targeted agricultural solutions, fostering scalable innovations for human and environmental health. These technology-driven approaches are fundamental to unlocking the full potential of holobiome science.

### 6.2. Conclusions

The holobiome represents a vast, interconnected network of microbial ecosystems spanning humans, animals, plants, and the environment. Its health is fundamental to the sustainability of life on Earth. This review highlights the transformative roles of probiotics, sustainable agricultural practices, and AI-driven innovations in restoring balance within the holobiome. Probiotics and microbial inoculants offer targeted solutions to enhance microbial diversity, improve nutrient cycling, and suppress pathogens, while sustainable agricultural practices reduce chemical inputs and promote ecosystem resilience. Furthermore, the integration of AI and systems biology is revolutionizing microbiome research, enabling precision interventions that offer scalable solutions to global challenges in health, agriculture, and environmental sustainability.

By fostering policies that support microbiome science, leveraging advanced technologies, and promoting interdisciplinary collaboration, society can address pressing issues such as climate change, food security, and public health disparities. The holobiome framework provides a roadmap for restoring microbial balance, emphasizing the potential for achieving global health and ecological sustainability.

By embracing a holobiome-centric approach, we can unlock the full potential of microbial ecosystems to create a healthier planet and a more equitable future for all. These efforts are essential for building a future where human and environmental health are harmonized, ensuring resilience and well-being for generations to come.

## Figures and Tables

**Figure 1 microorganisms-13-00514-f001:**
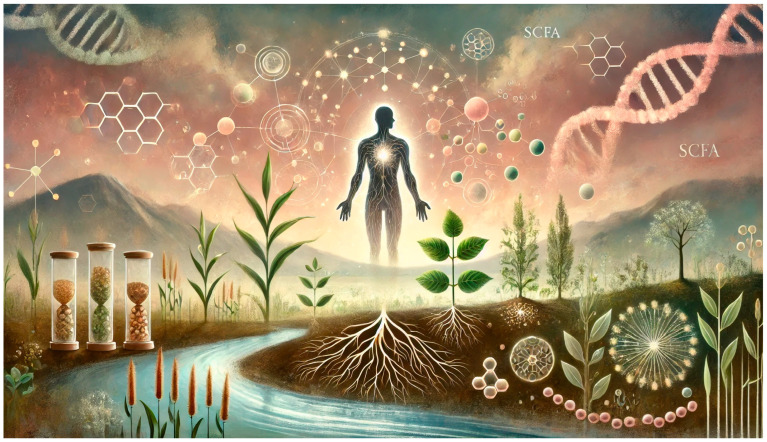
Holobiome interconnectedness: bridging microbial ecosystems, human health, and the environment.

**Figure 2 microorganisms-13-00514-f002:**
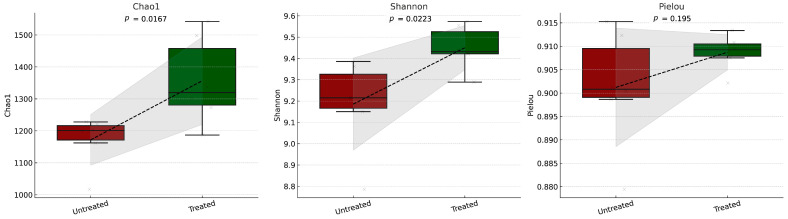
Comparison of alpha diversity indices between treated and untreated cohorts.

**Figure 3 microorganisms-13-00514-f003:**
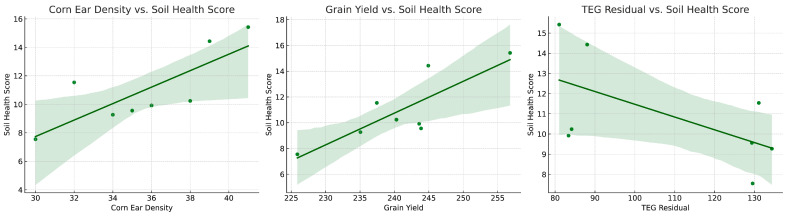
Effect of the microbial consortium on soil health and yield. This figure presents three scatter plots examining the relationship between soil health score and various agricultural metrics. Each plot displays individual data points as dots, representing specific measurements of the respective metric and corresponding soil health score. A solid green line, the regression line, illustrates the best-fit linear trend between the two variables, showing the general direction and strength of the relationship. Surrounding each regression line is a light green shaded area, the confidence interval, which indicates the range of uncertainty in the estimated relationship, providing a visual representation of the precision of the trend.

**Figure 4 microorganisms-13-00514-f004:**
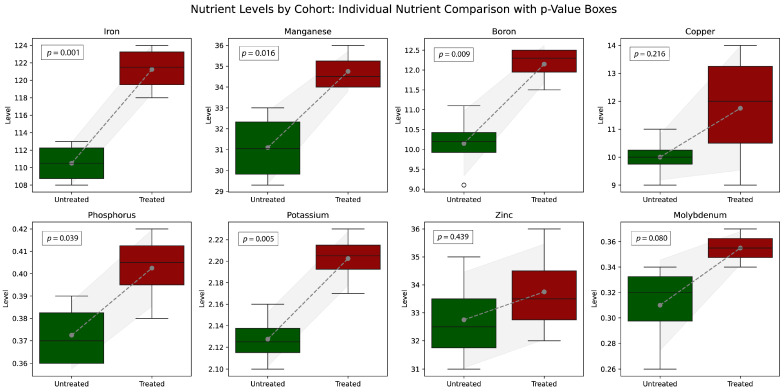
Impact of microbial soil inoculant on plant nutrient uptake: enhanced levels of essential micronutrients across treated and untreated cohorts. This figure presents box plots comparing the levels of various micronutrients (Iron, Manganese, Boron, Copper, Phosphorus, Potassium, Zinc, and Molybdenum) in untreated and treated soil samples. Green box plots represent the distribution of micronutrient values in the untreated soils, while burgundy box plots represent the distribution of micronutrient values in the treated soils. The dashed grey lines connect the medians of the untreated and treated groups, visually indicating the trend or change in median micronutrient levels between the two treatments. The light grey shaded area surrounding the trend line represents the confidence interval, providing an estimate of the uncertainty associated with the observed trend. The *p*-values shown in each plot indicate the statistical significance of the difference between the untreated and treated groups for each respective micronutrient.

**Figure 5 microorganisms-13-00514-f005:**
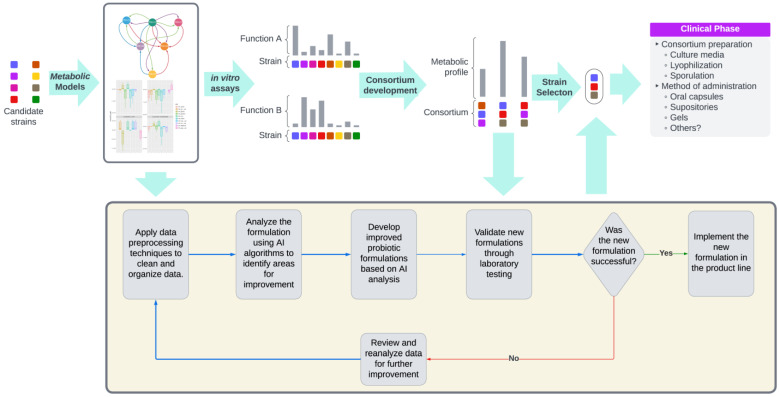
Integrative workflow for designing optimized microbial consortia using genomics, metabolic modeling, and AI. This figure illustrates the process of AI-driven probiotic formulation development. Initially, various bacterial strains (represented by different colored squares) are assessed for their individual production of a desired metabolite (indicated by the height of the gray bars). Subsequently, an AI algorithm analyzes this data to identify an optimized formula consisting of three strains (encapsulated within an oval). This selected strain composition demonstrates significantly enhanced metabolite production (*p* = 0.001), highlighting the efficacy of AI in developing superior probiotic formulations.

**Figure 6 microorganisms-13-00514-f006:**
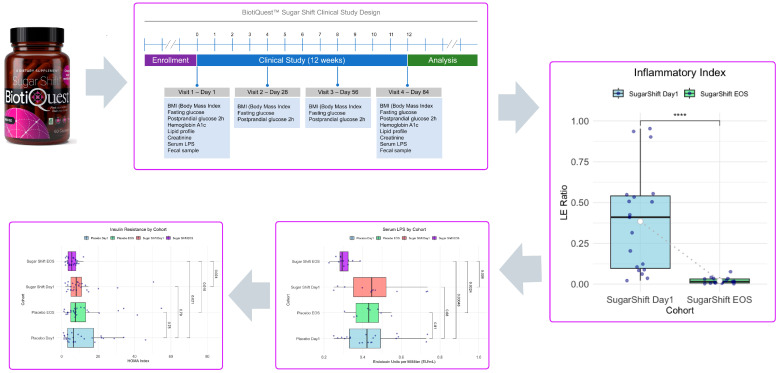
Congruence between microbiome results and clinical data. Treatment with the probiotic formulation induced microbiome transformations by reversing the inflammatory index, shifting the ratio of facultative anaerobes (FAs) to strict anaerobes (As) from FA-dominant to A-dominant by the end of the treatment. This shift was accompanied by a significant reduction in serum LPS levels and a corresponding decrease in insulin resistance, as measured by the HOMA index. **** indicate a statistical significant reduction at *p* < 0.0001 of the inflammatory index from baseline to end of study (EOS).

**Figure 7 microorganisms-13-00514-f007:**
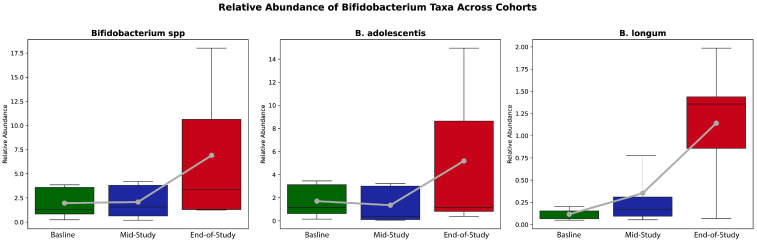
Relative abundance of *Bifidobacterium* taxa across cohorts over time. Changes in the relative abundance of *Bifidobacterium* spp., *B. adolescentis*, and *B. longum* in participants receiving fucoidan treatment at three time points: baseline, mid-study, and at end of study. Boxplots represent the interquartile range, with whiskers indicating variability outside the upper and lower quartiles, and lines connecting the mean values.

**Table 1 microorganisms-13-00514-t001:** Comparison of post-harvest soil health parameters between treated and untreated groups.

Parameter	Treated (Mean ± SD)	Untreated (Mean ± SD)
Soil Organic Carbon (mg/kg)	118.00 ± 5.62	148.00 ± 11.37
Soil Health Score	13.26 ± 2.33	8.79 ± 0.80
Microbial Organic Carbon (mg/kg)	80.48 ± 23.54	24.83 ± 5.18
CO_2_ Soil Respiration (mg/kg/day)	95.83 ± 30.34	35.88 ± 5.92
CEC (meq/100 g)	8.78 ± 0.22	7.13 ± 0.55
NO_3_-N (mg/kg)	38.15 ± 3.58	31.75 ± 4.43
NH_4_-N (mg/kg)	0.6 ± 0.08	1.68 ± 0.41
Potassium (K, mg/kg)	368.75 ± 11.14	289.75 ± 34.11
Organic Matter (LOI, %)	1.90 ± 0.07	1.70 ± 0.07

## Data Availability

The processed data and scripts used for statistical analysis and figure generation in this study will be deposited in a public GitHub repository to ensure transparency and reproducibility.

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
