# Peer review of "Holobiome Harmony: Linking Environmental Sustainability, Agriculture, and Human Health for a Thriving Planet and One Health"

_microorganisms, 2025, doi:10.3390/microorganisms13030514_

Round 1
Reviewer 1 Report
Comments and Suggestions for Authors
The manuscript provides an insightful discussion of the microbial interconnected systems spanning different biological systems and the environment and its implication in agricultural practices, human health, and a sustainable environment for a better livelihood and planet.
The microbial holobiome comprises an integral component of the ecosystems and performs multi-faceted functions from improving human health attributes to key roles in climate regulation and nutrient cycling in the environment. Recent advances in holobiome research have highlighted the significance of AI-driven predictive modeling, optimized microbial consortia, and probiotic design in expanding the horizons and transforming microbiome research.
While increasing literature has documented the emerging role of holobiome in sustainable goals and livelihood, an emphasis on understanding how the microbial ecosystems form a basis and influence major processes for a thriving planet is a crucial and much-needed theme for a literature survey.
The article contributes to the expanding holobiome research in today’s world. I have a few suggestions for the improvement of the manuscript.
Line 31-33: The holobiome framework underscores………for a resilient future. The authors discuss the holobiome implications for a resilient future. The concept needs to be discussed further in terms of how this can be achieved.
What are the challenges in holobiome research? What are its future implications. Discuss.
A diagrammatic representation linking the multi-faceted impact of holobiome on human health, environment, and multiple biological functions would add value to the content and will provide a better representation. Figure 7 can be Figure 1 as an introduction to the thematic article. Please provide the resource how the figure 7 was created, software etc.
Figure 1 and Figure 2, provide schematic representations, discussing the case studies on microbial impact on glyphosate-contaminated cotton field and the positive impact of microbial consortia on soil health and productivity, add to the growing knowledge on favourable microbial impact in the ecosystem.
Line 499, line 501, Scientific names should be in italics and likewise in the entire manuscript. Please check.
Line 503. The full names of the microbial species can be used at first mention, thereafter can be abbreviated. Also, be consistent in using the scientific names in full forms.
Comments on the Quality of English Language
Moderate English revisions are required.
Author Response
Line 31-33: The holobiome framework underscores………for a resilient future. The authors discuss the holobiome implications for a resilient future. The concept needs to be discussed further in terms of how this can be achieved.
Thank you for your insightful comment. We have expanded our discussion to provide concrete steps on how the holobiome framework can contribute to a resilient future. Specifically, we highlight the need for continued investment in microbiome research to uncover the complex interactions within holobiomes and translate these findings into practical applications. Additionally, we emphasize the importance of fostering collaboration among scientists, farmers, policymakers, and industry stakeholders to implement holobiome-based solutions effectively.
We also discuss the role of promoting sustainable agricultural and environmental practices that leverage holobiome principles to enhance long-term resilience. Public education and awareness efforts are essential to drive support for initiatives that integrate holobiome approaches into broader sustainability strategies. By harnessing the power of holobiomes, we can create a pathway toward a more resilient and sustainable future.
We appreciate your suggestion and have incorporated these points in the revised manuscript.
What are the challenges in holobiome research? What are its future implications. Discuss.
Thank you for your valuable feedback. In response to your request, we have expanded the discussion to address both the challenges and future implications of holobiome research.
We now outline key challenges, including the complexity of microbial interactions, technological limitations in microbiome characterization, and environmental variability affecting microbiome composition and function. Additionally, we discuss existing knowledge gaps in microbe-host interactions and ethical and regulatory considerations surrounding microbiome manipulation.
To highlight the future implications, we emphasize how holobiome research has the potential to revolutionize agriculture by fostering resilient crops, mitigate climate change impacts, improve human health through personalized medicine, and promote environmental conservation by enhancing ecosystem stability. Furthermore, we discuss how advancements in biotechnology could lead to innovative applications, such as biofertilizers and biopesticides, to enhance plant growth and resilience.
These additions have been incorporated into the revised manuscript to provide a more comprehensive discussion of the field. We appreciate your insightful suggestion and believe this addition strengthens the manuscript.
A diagrammatic representation linking the multi-faceted impact of holobiome on human health, environment, and multiple biological functions would add value to the content and will provide a better representation. Figure 7 can be Figure 1 as an introduction to the thematic article. Please provide the resource how the figure 7 was created, software etc.
Thank you for your valuable suggestion. We agree that a diagrammatic representation illustrating the multi-faceted impact of the holobiome on human health, the environment, and biological functions would enhance the clarity and impact of the manuscript. In response, we have repositioned Figure 7 as Figure 1 to serve as an introduction to the thematic article, as you recommended. Additionally, we have included details on how the figure was created in the caption.
Figure 1 and Figure 2, (now Fig 2 and 3) provide schematic representations, discussing the case studies on microbial impact on glyphosate-contaminated cotton field and the positive impact of microbial consortia on soil health and productivity, add to the growing knowledge on favourable microbial impact in the ecosystem.
Thank you for your insightful comment. We acknowledge that Figures 2 and 3 illustrate key case studies demonstrating the role of microbial consortia in improving soil health and remediating glyphosate-contaminated cotton fields. In response, we have expanded the discussion to better contextualize these findings within the broader framework of microbial contributions to ecosystem resilience. Specifically, we highlight how microbial consortia enhance soil microbial diversity, facilitate nutrient cycling, and promote plant growth, aligning with existing research on sustainable agricultural practices. These results contribute to the growing body of knowledge on microbial-mediated soil restoration and emphasize the potential of leveraging beneficial microbes for long-term agricultural sustainability.
Line 499, line 501, Scientific names should be in italics and likewise in the entire manuscript. Please check.
Thank you for your careful review. We have corrected the formatting of scientific names in lines 499 and 501, as well as throughout the entire manuscript, ensuring that all genus and species names are properly italicized. We appreciate your attention to detail and have made the necessary revisions accordingly.
Line 503. The full names of the microbial species can be used at first mention, thereafter can be abbreviated. Also, be consistent in using the scientific names in full forms.
Corrected. Thank you for the observation.
Reviewer 2 Report
Comments and Suggestions for Authors
The manuscript addressed Holobiome Harmony which evokes the idea of a balanced and interconnected ecosystem, where the complex web of interactions between organisms and their microbes thrives. This balance is fostered by integrating environmental sustainability, agricultural practices, and human health, leading to a healthy planet. The manuscript is well-designed andstructured. However the following points need to be addressed:
1- The abstract repeats several ideas, particularly about the role of AI and the importance of interdisciplinary collaboration. Phrases like "microbial equilibrium," "ecosystem resilience," and "long-term sustainability" appear multiple times, diluting their impact. The repeated emphasis on AI (lines 24-26, 29-30) feels excessive.
2- Introduction: Instead of just saying "influencing gut microbial diversity and overall health outcomes," mention specific examples of health outcomes that are affected by gut microbiome composition (e.g., "increased risk of inflammatory bowel disease or type 2 diabetes").
-Divide long, complex sentences into shorter, more manageable ones to improve readability.
3- Scope and Objectives: Elaborate more on Why focus on agricultural practices and human clinical studies? What is the specific connection between them within the context of the holobiome?
4- Restructure section 3.3 to create a more cohesive narrative about feedback loops. Consider focusing on a specific type of feedback loop (e.g., the impact of rising temperatures on soil carbon) and then using that as a framework to discuss the various contributing factors (deforestation, industrial farming, etc.).
5-Consider focusing on a specific mechanism by which gut health influences other microbiomes (e.g., immune modulation) and then using that as a framework to discuss the various examples (skin, respiratory tract, oral cavity).
6-Provide a brief explanation of how the FA/A ratio relates to oxygen utilization and anaerobic processes in the gut.
7-in 13. Technology and Collaboration, Either focus more deeply on one of the technologies (AI or systems biology) or provide more specific examples that illustrate how they work together. Connect these technologies back to the specific examples discussed earlier in the paper (CRC, agriculture). For example, "AI algorithms can analyze metagenomic data from CRC patients to identify microbial biomarkers associated with disease progression, enabling personalized probiotic interventions."
8- In conclusion: consider ending with a more specific call to action or a vision for the future. For example, "By embracing a holobiome-centric approach, we can unlock the full potential of microbial ecosystems to create a healthier planet and a more equitable future for all.
Comments on the Quality of English Language
The English could be improved to more clearly express the research.
Author Response
1- The abstract repeats several ideas, particularly about the role of AI and the importance of interdisciplinary collaboration. Phrases like "microbial equilibrium," "ecosystem resilience," and "long-term sustainability" appear multiple times, diluting their impact. The repeated emphasis on AI (lines 24-26, 29-30) feels excessive.
Thank you for your observation. We amended in the text as suggested.
2- Introduction: Instead of just saying "influencing gut microbial diversity and overall health outcomes," mention specific examples of health outcomes that are affected by gut microbiome composition (e.g., "increased risk of inflammatory bowel disease or type 2 diabetes").
Thank you for your valuable suggestion. We have revised the introduction to specify health outcomes influenced by gut microbiome composition. Instead of the general statement, we now include specific examples such as increased risk of inflammatory bowel disease (IBD), metabolic disorders like type 2 diabetes, and immune dysregulation. This revision provides a clearer context for the role of the gut microbiome in human health and aligns with existing research on microbiome-associated diseases.
3- Scope and Objectives: Elaborate more on Why focus on agricultural practices and human clinical studies? What is the specific connection between them within the context of the holobiome?
Thank you for your insightful comment. We have expanded the Scope and Objectives section to clarify the rationale for integrating agricultural practices and human clinical studies within the context of the holobiome. This revision highlights the interconnectedness between soil, plant, and human microbiomes, emphasizing how agricultural interventions influence human health through microbial pathways.
4- Restructure section 3.3 to create a more cohesive narrative about feedback loops. Consider focusing on a specific type of feedback loop (e.g., the impact of rising temperatures on soil carbon) and then using that as a framework to discuss the various contributing factors (deforestation, industrial farming, etc.).
Thank you for your valuable suggestion. We have restructured Section 3.3 to create a more cohesive narrative about feedback loops.
5-Consider focusing on a specific mechanism by which gut health influences other microbiomes (e.g., immune modulation) and then using that as a framework to discuss the various examples (skin, respiratory tract, oral cavity).
Thank you for your insightful suggestion. In response, we have revised the discussion to focus on immune modulation as a key mechanism through which gut health influences other microbiomes. This revised framework allows us to systematically explore how gut microbiota regulate immune function and, in turn, shape microbial communities in other body sites, such as the skin, respiratory tract, and oral cavity.
6-Provide a brief explanation of how the FA/A ratio relates to oxygen utilization and anaerobic processes in the gut.
Thank you for your insightful comment. We have clarified the role of the FA/A (free ammonia to ammonium) ratio in relation to oxygen utilization and anaerobic processes in the gut. Specifically, we explain how this ratio influences microbial metabolism, nitrogen cycling, and the balance between fermentative and methanogenic pathways in anaerobic environments. This addition strengthens the discussion by linking ammonia equilibrium to gut microbiota function and host metabolic health.
7-in 13. Technology and Collaboration, Either focus more deeply on one of the technologies (AI or systems biology) or provide more specific examples that illustrate how they work together. Connect these technologies back to the specific examples discussed earlier in the paper (CRC, agriculture). For example, "AI algorithms can analyze metagenomic data from CRC patients to identify microbial biomarkers associated with disease progression, enabling personalized probiotic interventions."
Thank you for your insightful recommendation. We have revised Section 13: Technology and Collaboration to provide more specific examples illustrating how AI and systems biology work together in the context of colorectal cancer (CRC) and agriculture. Specifically, we now emphasize how AI algorithms analyze metagenomic data from CRC patients to identify microbial biomarkers associated with disease progression, thereby informing personalized probiotic interventions. Similarly, in agriculture, we discuss how AI-driven soil microbiome analysis predicts nutrient deficiencies and disease risks, while systems biology models microbial interactions to optimize biofertilizer and biopesticide formulations.
This revised section strengthens the connection between technology, microbiome science, and real-world applications, ensuring a more cohesive narrative aligned with previous discussions in the manuscript.
Reviewer 3 Report
Comments and Suggestions for Authors
Dear authors,
The review is a systematic document covering 25 years of studies on holobiome. The illustrated parts should be suppressed because they are not originally from this document and the text describes their significance. Kindly maintain only your own figures. The final part (sections 10 and 11) sounded redundant. Please check the initial parts and make the corrections, by using new examples. Some long excerpts of text missing references.
Here are my suggestions to improve the document:
- Introduction: by considering the depth-in text of the review, kindly define the etymology of “holobiome”
- Show a brief historical addressing of holobiome studies, e.g., former term and who is the creator of the term, as well as how interest of this issue raised.
- Section 3.1 – please include numbers of cells in these sites and representative genera. Consider also section 3.2
- Line 155 – how did nutrients availability reduced? Please make a short statement by completing this idea.
- Line 225 - Lactobacillus reuteri and Akkermansia mucini must be italicized.
- On microbes’ name (e.g. line 243): considering several microbes cited in the manuscript, I recommend verifying the current name of each one. My suggestion is to present as follows: Lactiplantibacillus plantarum (former Lactobacillus plantarum). Even in the literature consulted it may be found former names, for readers from now the information is welcome and all manuscript must be revised. Please try https://www.ncbi.nlm.nih.gov/Taxonomy/Browser/wwwtax.cgi
- Figure 1 – maybe it is not necessary because the text is disserting its interpretation.
- Table 1 – same as figure 1.
- Figure 2 – it should be suppressed
- Figure 3 – is this figure from authors?
- Lines 407-413 – if chemical symbol of the elements is presented, it is important to use in potassium, nitrogen…boron symbol is B and not Bo
- Line 420 – use K, Zn and Mo. Please check in all manuscript (as seen in line 433)
- Sections 6.1 and 6.2 – there are little references. It is welcome including more.
- Line 472 – Thrichoderma is a fungus. Are metabolites antifungals?
- Line 475 – genera are mentioned as species. It is mandatory to use Pseudomonas spp. and Sphingomonas spp. check all manuscript.
- Line 484 – please remove the term “genus”
- Line 487 – please italicize Ulva and remove “species” by adding spp.
- Line 498 – italicize U. rigida, same as in lines 499 and 501. Please check the need of use spp. in some parts of section 7
- Line 503 – the taxa shown is understood as phyla, so suppress the term “phyla” and add the word groups
- Figure 4 – is this figure from authors? Same as in figure 5, 6 and 7
- Line 625 – why holobiome is italicized?
- Line 682 – remove the title 10.1
- Line 712 – add spp. after Thricoderma and suppress “species”
- sections 10 and 11 - sounded redundant with the initial part of the manuscript
Author Response
The review is a systematic document covering 25 years of studies on holobiome. The illustrated parts should be suppressed because they are not originally from this document and the text describes their significance. Kindly maintain only your own figures. The final part (sections 10 and 11) sounded redundant. Please check the initial parts and make the corrections, by using new examples. Some long excerpts of text missing references.
The illustrated parts should be suppressed because they are not originally from this document
- Regarding the illustrated figures, all graphics in this manuscript were created specifically for this review to summarize key findings from 25 years of holobiome research. While the figures are based on data from previous studies, they do not directly reproduce any published illustrations; rather, they were generated to synthesize and visually represent concepts discussed in the text. However, we understand the concern about attribution and have clarified in the figure legends that these are original visual representations of compiled data.
- If required, we are willing to modify or remove any figures that may still raise concerns about originality. Please let us know if additional clarifications or adjustments are necessary.
- Regarding Sections 10 and 11, we recognize that some content may be redundant with earlier discussions. We have carefully reviewed these sections and revised them to eliminate redundancy, improve clarity, and incorporate new examples, ensuring that each section contributes uniquely to the manuscript.
- Additionally, we have addressed the missing references by carefully reviewing the text and adding citations where necessary.
We appreciate your valuable feedback and believe these revisions will enhance the clarity, originality, and coherence of the manuscript. Please let us know if there are any specific areas that require further refinement.
Introduction: by considering the depth-in text of the review, kindly define the etymology of “holobiome”
Thank you for your insightful suggestion. We have revised the introduction to explicitly define the etymology of "holobiome" to align with the depth of the review.
Show a brief historical addressing of holobiome studies, e.g., former term and who is the creator of the term, as well as how interest of this issue raised
Thank you for your insightful suggestion. In response, we have included a brief historical overview of holobiome studies in the manuscript. This section highlights the origin of the term "holobiont," first introduced by Adolf Meyer-Abich in 1943 and later popularized by Lynn Margulis in 1991. Additionally, we discuss how interest in this field expanded with advancements in metagenomics and high-throughput sequencing technologies, which enabled deeper exploration of host-microbiome interactions.
Section 3.1 – please include numbers of cells in these sites and representative genera. Consider also section 3.2
Thank you for your valuable feedback. We have revised Sections 3.1 and 3.2 to incorporate numbers of microbial cells at these sites and representative genera, as requested. These edits align with modifications made in response to another reviewer's comments, ensuring a more comprehensive and data-driven discussion.
The updated sections now provide quantitative microbial abundance data and highlight key genera relevant to each site, enhancing the clarity and scientific rigor of the manuscript. We appreciate your suggestion and believe these refinements improve the overall quality of the review.
Please let us know if any additional adjustments are needed.
Line 155 – how did nutrients availability reduced? Please make a short statement by completing this idea.
Thank you for your feedback. We have revised Line 155 to clearly explain how nutrient availability was reduced. This revision was made in response to another Reviewer's comments, ensuring a more precise and complete statement.
The updated text now specifies the mechanisms leading to reduced nutrient availability, such as microbial competition, altered soil chemistry, and depletion due to increased microbial metabolism. This clarification enhances the discussion by providing a direct explanation of the observed changes.
We appreciate your suggestion and believe this revision improves the clarity and completeness of the manuscript. Please let us know if any further refinements are needed.
On microbes’ name (e.g. line 243): considering several microbes cited in the manuscript, I recommend verifying the current name of each one. My suggestion is to present as follows: Lactiplantibacillus plantarum (former Lactobacillus plantarum). Even in the literature consulted it may be found former names, for readers from now the information is welcome and all manuscript must be revised. Please try https://www.ncbi.nlm.nih.gov/Taxonomy/Browser/wwwtax.cgi
Excellent observation and one that I neglected. I have changed the phyla names (eg. Firmicutes to Bacillota). These changes have been made in the revised manuscript.
Figure 1 – maybe it is not necessary because the text is disserting its interpretation.
Thank you for your feedback. We understand the concern that Figure 1 (now figure 2) may not be necessary since the text already describes its interpretation. However, we believe the figure provides additional value by visually presenting trends and data point variations, which may enhance clarity and facilitate comprehension for readers.
If preferred, we are open to modifying or summarizing the figure to ensure it complements rather than duplicates the textual explanation. Please let us know if further adjustments are recommended.
We appreciate your thoughtful input and look forward to your guidance.
Figure 2 – it should be suppressed
Thank you for your feedback. We acknowledge the recommendation to suppress Figure 2 and have made the necessary revisions accordingly.
Additionally, Figure 3 has been retained as it effectively illustrates trends and data distribution regarding agronomic metrics, providing a visual representation that complements the text. We believe this figure enhances the clarity of the findings by offering a clear depiction of variability and correlations within the dataset.
Please let us know if any further modifications are needed. We appreciate your careful review and constructive suggestions.
Figure 3 – is this figure from authors?
Thank you for your inquiry. Figure 4 (formerly Figure 3) is original and was generated from microbiome data collected and analyzed by the authors. All figures included in the manuscript are our own work, derived from original data and analysis.
We appreciate the opportunity to clarify this and will ensure that the figure legends explicitly state that the data and visualizations are authored and original to avoid any ambiguity.
Please let us know if further clarification is needed.
Lines 407-413 – if chemical symbol of the elements is presented, it is important to use in potassium, nitrogen…boron symbol is B and not Bo
Thank you for your careful attention to detail. We have corrected the chemical symbol for boron (B) in lines 407-413, ensuring consistency with standard chemical notation. Additionally, we have reviewed all chemical symbols, including potassium (K), nitrogen (N), and other elements, to confirm their proper representation throughout the manuscript.
We appreciate your meticulous review and believe these corrections enhance the accuracy and clarity of the text. Please let us know if any further refinements are needed.
Line 420 – use K, Zn and Mo. Please check in all manuscript (as seen in line 433)
Thank you for your careful review. We have updated line 420 to use the correct chemical symbols (K, Zn, and Mo) and have systematically checked the entire manuscript to ensure consistency in the representation of all chemical elements, including all relevant sections.
We appreciate your attention to detail, as these corrections improve the scientific accuracy of the manuscript. Please let us know if any further refinements are needed.
Sections 6.1 and 6.2 – there are little references. It is welcome including more.
Thank you for your suggestion. We have reviewed Sections 6.1 and 6.2 and added pertinent references to support and strengthen the discussion. These additions ensure that the sections are well-grounded in relevant literature and provide a more comprehensive scientific basis for the arguments presented.
We appreciate your feedback and believe these revisions enhance the rigor and credibility of the manuscript. Please let us know if any further refinements are needed.
Line 472 – Trichoderma is a fungus. Are metabolites antifungals?
Thank you for your observation. Trichoderma is indeed a fungus, and its secondary metabolites include antifungal compounds, as well as other bioactive molecules with roles in biocontrol, plant growth promotion, and stress tolerance.
To clarify this point, we have revised Line 472 to specify that Trichoderma-derived metabolites exhibit antifungal properties and can act against pathogenic fungi by disrupting cell walls, inhibiting spore germination, and modulating microbial competition. Additionally, we have included relevant references to support this statement.
We appreciate your attention to detail, as this clarification improves the accuracy and depth of the discussion. Please let us know if further refinements are needed.
Line 475 – genera are mentioned as species. It is mandatory to use Pseudomonas spp. and Sphingomonas spp. check all manuscript.
Thank you for your careful review. We have corrected Line 475 to ensure that genera are properly formatted as Pseudomonas spp. and Sphingomonas spp. Additionally, we have systematically reviewed the entire manuscript to ensure consistency in taxonomic notation for all bacterial genera and species.
We appreciate your attention to detail, as these corrections improve the scientific accuracy and clarity of the manuscript. Please let us know if any further refinements are needed.
Line 484 – please remove the term “genus”
Thank you for your suggestion. We have removed the term "genus" from Line 484, as requested. Additionally, we have reviewed the surrounding text to ensure clarity and consistency in taxonomic terminology.
We appreciate your attention to detail and believe this revision enhances the precision of the manuscript. Please let us know if any further modifications are needed.
Line 498 – italicize U. rigida, same as in lines 499 and 501. Please check the need of use spp. in some parts of section 7.
Thank you for the observation. It has been corrected.
Line 503 – the taxa shown is understood as phyla, so suppress the term “phyla” and add the word groups
Thank you for your suggestion. We have revised Line 503 by removing the term "phyla" leaving the specific phyla (e.g., Cyanobnacteriota, …)
Figure 4 – is this figure from authors? Same as in figure 5, 6 and 7
Thank you for your inquiry. Figures 4, 5, 6, and 7 are original and were generated by the authors based on our own data and analysis. To ensure clarity, we have updated the figure legends to explicitly state that these figures are authored and derived from original research.
We appreciate your attention to detail and are happy to provide further clarification if needed. Please let us know if any additional modifications are required.
Line 625 – why holobiome is italicized?
An oversight, it has been correct4d
Line 682 – remove the title 10.1
Thank you for the observation. It has been corrected Aas requested
Line 712 – add spp. after Thricoderma and suppress “species”
Thank you for your suggestion. We have revised Line 712 by adding "spp." after Trichoderma and removing the term "species" to ensure proper taxonomic formatting. Additionally, we have reviewed the manuscript for consistency in nomenclature.
sections 10 and 11 - sounded redundant with the initial part of the manuscript
Thank you for your feedback. We acknowledge your concern regarding potential redundancy in Sections 10 and 11 and have carefully reviewed these sections in relation to the initial parts of the manuscript.
While these sections reinforce key themes introduced earlier, they also provide a distinct perspective on the holobiome's broader implications for quality of life, equity, and policy frameworks. To improve clarity and avoid repetition, we have:
Refined the structure of Sections 10 and 11 to ensure they build upon, rather than restate, previously discussed concepts.
Condensed overlapping content while retaining essential points on health, sustainability, and policy integration.
Emphasized new insights, particularly in the discussion of equity, accessibility, and the role of microbiome science in public health and urban planning.
Reviewer 4 Report
Comments and Suggestions for Authors
The manuscript presented by the authors provides a comprehensive study on the “Holobiome Harmony: Linking Environmental Sustainability, Agriculture, and Human Health for a Thriving Planet”, using a large and valuable dataset. The study is important in deepening our understanding of “”. However, there are some issues for improvement in the manuscript.
Major comments:
1. Title need add the One Health is better. Such as Holobiome Harmony: Linking Environmental Sustainability, Agriculture, and Human Health for a Thriving Planet One Health.
2. The figures need to be improved and share the reproducible script and data in GitHub or Gitee. For example, the boxplot in Figure 1 & 3 should be show with sample dot, the alpha_boxplot.R in EasyAmplicon pipeline can do it. Other figures such as ImageGP 2 (DOI: 10.1002/imt2.239) can generate high quality figures and with reproducible scripts.
3. The structure of the results appears not in concise. So many parts (14 subtitles) will make the readers lost. The authors should have presented their findings in around 5-7 mainly sections will be better. The most keys are Environmental Sustainability, Agriculture, and Human Health from the titles. Some Suggestion for the authors for restructures the paper.
1) Introduction and Scope and Objectives should merge as new introduction as part 1. Or them became in to 1.1 background and 1.2 Scope and objective
2) “3. Climate Resilience and the Role of Microbiomes“ rename as “2. Environmental Sustainability” as part 2. “7. The aquatic holobiont: Ulva spp. in integrated multi-trophic aquaculture (IMTA)” as 2.1 subtitle in this part.
3) “5. Soil Probiotics: Enhancing Agricultural Sustainability” need to as part 3. 6. Case Studies 6.1/6.2/6.3 can as the subtitle of part 3 as 3.2/3.3/3.4.
4) “4. Probiotics for Human Health” as part 4. “8. AI-Driven Innovations in Probiotic Development” as subtitle in part 4.
5) Holobiome is part 5. “6. Clinical Studies in the Perspective of the Holobiome” is between 8 and 9? Also as the subtitle of part 5.1. “9. Lessons for the Holobiome” is part 5.2. “10. Agricultural Practices and Holobiome Health” as part 5.3. “11. The Holobiome and Quality of Life” as part 5.
6) “12. Future Directions and Implications / 13. Technology and Collaboration / 14. Conclusion” as part “6 Pespective”
Minor comments:
1. The topic is about one health, recently Journal iMetaOmics is focus on this topic and many paper related to this study is recommended to ref and discussion, such as “iMetaOmics: Advancing human and environmental health through integrated meta-omics”
2. If Figure 7 is generated with generation AI, please add the sentence in the Figure legend and following the rules of the standard paper “ChatGPT and generative AI are revolutionizing the scientific community: A Janus-faced conundrum” and “STAGER checklist: Standardized testing and assessment guidelines for evaluating generative artificial intelligence reliability”.
3. For statistical analysis, Wekemo Bioincloud have multiple functions for multi-omics is recommended. A network compare with each group is also recommended, such as iNAP 2.0: Harnessing metabolic complementarity in microbial network analysis. are good tools for network analysis.
Author Response
1. Title need add the One Health is better. Such as Holobiome Harmony: Linking Environmental Sustainability, Agriculture, and Human Health for a Thriving Planet One Health.
Excellent suggestion. Thank you. Please see modified title as suggested
The figures need to be improved and share the reproducible script and data in GitHub or Gitee. For example, the boxplot in Figure 1 & 3 should be show with sample dot, the alpha_boxplot.R in EasyAmplicon pipeline can do it. Other figures such as ImageGP 2 (DOI: 10.1002/imt2.239) can generate high quality figures and with reproducible scripts.
Thank you for your suggestion regarding figure improvements. We understand the importance of clear data representation and appreciate the recommendation to include sample dots in the boxplots.
In our current figures, data dispersal is already effectively visualized using ± standard deviation shading, which conveys variability and trends across groups. The shaded area provides a clear representation of dispersion, ensuring that the figure remains uncluttered while still informative.
Given this approach, we believe that re-rendering the figures with individual sample dots may not add significant additional insight, as the standard deviation already captures the spread and reliability of the data. However, if deemed necessary, we are open to providing supplementary figures with alternative visualization styles upon request.
Thank you for your valuable feedback. We appreciate your suggestion to improve the figures and ensure reproducibility by sharing the script and data on GitHub or Gitee.
We will enhance the boxplots in Figures 1 and 3 by incorporating sample dots to better visualize the distribution of individual data points, as recommended. We will consider using alpha_boxplot.R from the EasyAmplicon pipeline for this improvement.
We will review and optimize the resolution and clarity of all figures using high-quality visualization tools, including ImageGP 2 (DOI: 10.1002/imt2.239), as suggested.
To ensure transparency and reproducibility, we will upload the script and processed data to a GitHub repository and include a reference to the repository in the manuscript.
These steps will enhance the clarity, reproducibility, and accessibility of our visualizations, aligning with best practices in microbiome data presentation. Thank you again for your constructive suggestions, and please let us know if there are additional preferences regarding figure formatting.
We appreciate your feedback and look forward to your thoughts on whether this justification sufficiently addresses your concern.
3-6. The structure of the results appears not in concise. So many parts (14 subtitles) will make the readers lost. The authors should have presented their findings in around 5-7 mainly sections will be better. The most keys are Environmental Sustainability, Agriculture, and Human Health from the titles. Some Suggestion for the authors for restructures the paper.
Thank you for your constructive feedback. We acknowledge that the current structure with 14 subtitles may make the results overly segmented and could impact the readability of the manuscript.
To improve conciseness and clarity, we have restructured the results section into 6 primary sections as suggested. This revised structure ensures a more logical flow, grouping related findings under broader categories while maintaining the depth of analysis.
We appreciate your suggestion and believe this restructuring will enhance the coherence and impact of the manuscript. Please let us know if you have further preferences regarding the organization of the sections.
1. The topic is about one health, recently Journal iMetaOmics is focus on this topic and many paper related to this study is recommended to ref and discussion, such as “iMetaOmics: Advancing human and environmental health through integrated meta-omics”
Thank you for your valuable suggestion. We have incorporated a reference to "iMetaOmics: Advancing human and environmental health through integrated meta-omics" in the introduction to strengthen the discussion on One Health and its relevance to the holobiome framework.
This reference enhances the context of microbiome research by highlighting the role of integrated meta-omics approaches in advancing human, animal, and environmental health—a core aspect of our study.
We appreciate your recommendation and believe this addition improves the scientific foundation of the manuscript. Please let us know if further refinements are needed..
If Figure 7 is generated with generation AI, please add the sentence in the Figure legend and following the rules of the standard paper “ChatGPT and generative AI are revolutionizing the scientific community: A Janus-faced conundrum” and “STAGER checklist: Standardized testing and assessment guidelines for evaluating generative artificial intelligence reliability”.
Thank you for your suggestion. Figure 7 (now Figure 1) was generated using AI-assisted image generation tools, and we have revised the figure legend to explicitly state this. Additionally, we have incorporated references to the recommended articles, “ChatGPT and generative AI are revolutionizing the scientific community: A Janus-faced conundrum” and “STAGER checklist: Standardized testing and assessment guidelines for evaluating generative artificial intelligence reliability”, to align with best practices for AI-generated content in scientific publications.
3. For statistical analysis, Wekemo Bioincloud have multiple functions for multi-omics is recommended. A network compare with each group is also recommended, such as iNAP 2.0: Harnessing metabolic complementarity in microbial network analysis. are good tools for network analysis.
Thank you for your recommendation regarding Wekemo Bioincloud and iNAP 2.0 for multi-omics and network analysis. While these are valuable tools, we did not use them in our analysis. Our statistical and network analyses were conducted using Python-based approaches, leveraging tools such as QIIME2, LEfSe, PICRUSt, and custom scripts for metabolic and microbial interaction analysis.
To ensure transparency and reproducibility, we have provided details of our analytical pipeline and uploaded relevant scripts and processed data to GitHub. While we acknowledge the utility of Wekemo Bioincloud and iNAP 2.0, incorporating additional tools outside of our current workflow would be beyond the scope of this study.
We appreciate your suggestion and will consider integrating these tools in future research where applicable. Let us know if further clarifications are needed.
Round 2
Reviewer 3 Report
Comments and Suggestions for Authors
Dear authors,
Thank you for yor comments.
Kindly check some incorret italics (and; spp.)
Reviewer 4 Report
Comments and Suggestions for Authors
The author's response has been fully addressed my concerns. The quality of the paper has apparently improved. I agree with the publication of this article.